# BeaconKV: Key-Value Cache Compression Guided by Beacon Queries for Efficient Large Reasoning Model Inference

**Janghyeon Kim** [1]  **Minsoo Kim** [1]  **Kyuhong Shim** [2]  **Jungwook Choi** [1]

## Abstract

Large Reasoning Models (LRMs) achieve superior problem-solving through extended Chain-of-Thought (CoT) generation, but the resulting key-value (KV) cache grows linearly with sequence length and creates severe memory bottlenecks—often exceeding GPU capacity for long reasoning traces. Existing KV cache compression methods rely on recent queries to estimate future token importance, implicitly assuming these serve as reliable proxies for future attention patterns. We demonstrate that this assumption fails in long-horizon reasoning: certain decoding steps generate Thought Revisiting Tokens (TRT) that re-attend to distant previous context, such as task-solving plans formulated early in the trace. Through systematic analysis, we discover that queries corresponding to the TRT cluster into a small number of similarity groups in the embedding space. Based on this insight, we propose BeaconKV, a training-free KV cache compression method that maintains beacon queries—compact representatives for each global query cluster—to anticipate which KV pairs will be revisited without storing the entire query history. Across four open-source LRMs and diverse reasoning benchmarks, BeaconKV generally outperforms existing compression methods, achieving up to $5.8\times$ memory reduction while nearly preserving full cache accuracy and improving throughput by over $4.3\times$.

## 1. Introduction

Large Reasoning Models (LRMs) (Guo et al., 2025; OpenAI, 2024; Anthropic, 2025) have emerged as a transformative paradigm in artificial intelligence, demonstrating remarkable capabilities across tasks that demand sophisticated logical thinking, including mathematics, science, and coding. Models such as o1 (OpenAI, 2024), Claude Opus 4.5 (Anthropic, 2025), GPT-5.2 (OpenAI, 2025), and Gemini 3 Pro (Google, 2025) have achieved unprecedented performance on challenging reasoning benchmarks, while open-source alternatives like DeepSeek-R1 (Guo et al., 2025) and Qwen3 (Yang et al., 2025) have rapidly expanded access to these powerful reasoning capabilities. The potential of LRMs to tackle complex, multi-step problems positions them as foundational technology for next-generation AI applications.

The superior reasoning capabilities of LRMs fundamentally stem from inference-time scaling through extended Chain-of-Thought (CoT) generation (Wei et al., 2022). Unlike conventional language models that produce concise outputs, LRMs deliberately generate lengthy reasoning traces—often spanning tens of thousands of tokens—to systematically develop task-solving strategies before arriving at final answers. While this prolonged generation is essential for reasoning quality, it introduces severe computational challenges. In Transformer architectures (Vaswani et al., 2017), autoregressive decoding requires caching key-value (KV) pairs for all previously generated tokens, leading to a KV cache that grows linearly with sequence length. For instance, when Qwen3-4B generates 32K tokens with a batch size of 16, the KV cache alone can exceed 77GB—bringing a single 80GB GPU close to its memory limit. This memory bottleneck fundamentally limits the practical deployment of LRMs under constrained GPU resources.

Recent efforts have attempted to address this challenge through KV cache compression methods tailored for LRMs. RPC (Song et al., 2025) compresses KV caches by scoring token importance based on attention weights computed from recent queries, while R-KV (Cai et al., 2025) augments this approach by incorporating redundancy scores based on key similarity. However, these methods share a fundamental limitation rooted in a critical distinction between LRM inference and conventional long-context processing: in LRMs, tokens are generated on-the-fly during the reasoning process, making it inherently difficult to predict which tokens will become important in subsequent decoding steps. Existing methods attempt to approximate future importance

[1]Hanyang University [2]Sungkyunkwan University. Correspondence to: Jungwook Choi <choij@hanyang.ac.kr>.

*Proceedings of the 43rd International Conference on Machine Learning*, Seoul, South Korea. PMLR 306, 2026. Copyright 2026 by the author(s).

by relying on recent queries at eviction time, implicitly assuming that these queries serve as reliable proxies for future attention patterns. As we demonstrate in this work, this assumption fails to capture a crucial phenomenon in long-horizon reasoning, leading to premature eviction of tokens that prove essential later in the reasoning trace.

In this paper, we introduce BeaconKV, a training-free KV cache compression method motivated by a novel observation we term Thought Revisiting Tokens (TRT). Through systematic analysis of attention dynamics during LRM inference, we discover that certain decoding steps generate tokens that re-attend to distant previous context—such as task-solving plans formulated early in the reasoning process—to maintain global coherence throughout extended reasoning (Figure 1 and 2). We observe that queries can be categorized into two distinct types: local queries, which predominantly attend to nearby keys, and global queries, which correspond to TRT and attend to distant keys across the reasoning trace.

Crucially, we find that global queries are not randomly distributed but instead cluster into a small number of similarity groups in the query embedding space (Figure 4). This geometric structure suggests that the diverse set of global queries can be effectively represented by a compact set of *beacon queries*—representative queries for each cluster. By maintaining beacon queries alongside recent queries, BeaconKV can anticipate which KV pairs will be revisited by future global queries without storing the entire query history. To enable memory-efficient beacon query identification during inference, we propose Continual Farthest Point Sampling (FPS), an online algorithm that progressively selects geometrically diverse queries while maintaining a bounded memory footprint.

We evaluate BeaconKV across four open-source LRMs (R1-Distill-Qwen-7B, R1-Distill-Llama-8B, Qwen3-4B, and Qwen3-14B) on diverse reasoning benchmarks, including AIME24, MATH-500, LiveCodeBench, and GPQA-Diamond. Experimental results demonstrate that BeaconKV generally outperforms other KV cache compression methods, including RPC and R-KV, across a broad range of budget configurations. In terms of accuracy, BeaconKV achieves gains of up to 31.7 percentage points over existing methods. Under aggressive compression, BeaconKV reduces peak GPU memory usage by up to $5.8\times$ while nearly preserving accuracy comparable to full KV inference, and achieves throughput improvements of over $4.3\times$ compared to the uncompressed baseline. These results establish BeaconKV as an effective solution for deploying LRMs under constrained memory budgets.

## 2. Background

### 2.1. Attention Formulation with KV Caching

We review the attention formulation in Transformers (Vaswani et al., 2017) and the use of key-value (KV) caching during autoregressive decoding. Given an input sequence with hidden states $X \in \mathbb{R}^{N \times d}$, where $N$ denotes the sequence length and $d$ the hidden dimension, a single attention head projects each token into queries, keys, and values as

$$Q = XW_Q, \quad K = XW_K, \quad V = XW_V, \quad (1)$$

where $W_Q, W_K, W_V \in \mathbb{R}^{d \times d}$. Let $q_i, k_j, v_j$ denote the query, key, and value corresponding to tokens $i$ and $j$, respectively.

The attention output is then computed as

$$O^{\text{attn}} = \text{Softmax}\Big(QK^\top/\sqrt{d} + M\Big) V, \quad (2)$$

where $M$ denotes the causal attention mask.

In autoregressive decoding, the key-value pairs of previously generated tokens are stored in a KV cache. Let $(K, V)$ denote the cached keys and values accumulated up to decoding step $t$. At step $t+1$, only the query, key, and value of the new token are computed, and attention is evaluated as

$$o^{\text{attn}}_{t+1} = \text{Softmax}\Big(q_{t+1}\big[K\|k_{t+1}\big]^\top/\sqrt{d} + M\Big)\big[V\|v_{t+1}\big].$$
$$(3)$$

The newly generated KV pair $(k_{t+1}, v_{t+1})$ is appended to the KV cache, causing the cache size to grow linearly with the decoding length, which becomes a major memory bottleneck in long-context and reasoning tasks.

### 2.2. Attention-Based KV Scoring with Recent Queries

To mitigate the memory overhead caused by KV cache growth during long decoding, recent KV cache eviction methods (Li et al., 2024; Song et al., 2025; Cai et al., 2025) compress the cache by assigning importance scores to cached KV pairs and evicting those deemed less relevant. A common design choice in these methods is to estimate KV importance using attention weights induced by a small set of recently generated queries, motivated by the intuition that recent queries provide informative signals for near-future decoding.

Concretely, let $q_\tau \in \mathbb{R}^d$ denote the query at decoding step $\tau$ and $k_j \in \mathbb{R}^d$ denote a cached key at index $j$. The attention weight from $q_\tau$ to $k_j$ is computed using scaled dot-product attention:

$$w_{\tau,j} = \frac{\exp\Big(q_\tau^\top k_j/\sqrt{d}\Big)}{\sum_l \exp\Big(q_\tau^\top k_l/\sqrt{d}\Big)}. \quad (4)$$

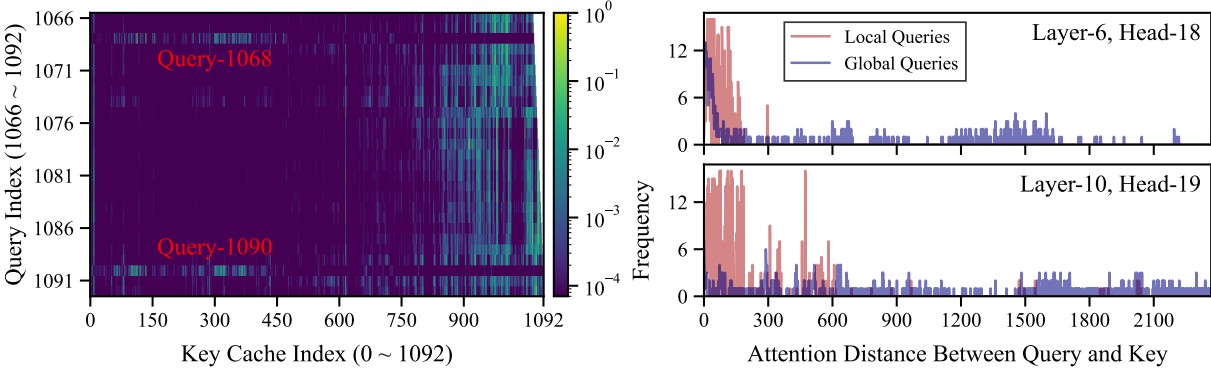

*Figure 1.* (a) Attention weight distribution at layer 18, head 16 and (b) attention distance between query and key (AIME24 sample-0 on R1-Distill-Qwen-7B).

Recent attention-based KV scoring methods ([Li et al., 2024](#); [Song et al., 2025](#); [Cai et al., 2025](#)) aggregate such attention weights over a set of *observation queries*, denoted as $Q_{\text{obs}}$. Observation queries are defined as a set of queries from the most recent decoding steps, where the window size $N_{\text{obs}}$ controls how many recent queries are used to estimate KV importance. Formally, when KV eviction is triggered at decoding step $t$, the observation query set is defined as

$$Q_{\text{obs}} := \{q_\tau \mid \tau \in \mathcal{T}_{\text{obs}}\}, \quad \mathcal{T}_{\text{obs}} = \{t - N_{\text{obs}} + 1, \ldots, t\}. \tag{5}$$

Given the observation query set $Q_{\text{obs}}$, the importance score $s_j$ of a cached key $k_j$ is computed by aggregating the corresponding attention weights induced by queries in $Q_{\text{obs}}$, either by taking the maximum or the average across the observation window.

$$s_j^{\max} = \max_{\tau \in \mathcal{T}_{\text{obs}}} w_{\tau,j}, \quad s_j^{\text{mean}} = \frac{1}{N_{\text{obs}}} \sum_{\tau \in \mathcal{T}_{\text{obs}}} w_{\tau,j}. \tag{6}$$

Using the resulting importance scores $\{s_j\}_{j=1}^L$ and a KV cache budget $B_{\text{KV}}$, KV cache eviction is performed by retaining the top-$B_{\text{KV}}$ KV pairs. Let $\mathcal{I} \subset \{1, \ldots, L\}$ denote the index set of the top-$B_{\text{KV}}$ scores:

$$\mathcal{I} = \text{TopK}\big(\{s_j\}_{j=1}^L, B_{\text{KV}}\big). \tag{7}$$

The evicted KV cache is then obtained by indexing along the sequence dimension as

$$\hat{K} \leftarrow K[:,\mathcal{I},:], \qquad \hat{V} \leftarrow V[:,\mathcal{I},:]. \tag{8}$$

## 3. Observation

In this section, we analyze attention patterns during extended reasoning and identify a recurring phenomenon that we term Thought Revisiting Tokens (TRT). We demonstrate that existing KV cache compression methods fundamentally fail to capture this phenomenon and reveal that queries that induce TRT exhibit a geometric structure that enables a compact representation using a small set of beacon queries.

### 3.1. Thought Revisiting Tokens in Reasoning Trace

We begin by categorizing queries generated during LRM inference based on their attention patterns. We define local queries as those that predominantly attend to keys in their immediate neighborhood, reflecting the typical pattern where each token focuses on recent context for local coherence. In contrast, we define global queries as those that attend to keys at substantially distant positions, spanning across the reasoning trace to access earlier context. Tokens whose queries exhibit this global attention pattern—revisiting previously established reasoning context, such as problem statements or task-solving plans—are referred to as Thought Revisiting Tokens (TRT).

Figure [1](#)(a) visualizes the attention weight distribution between queries and cached keys for a representative attention head during LRM inference. The majority of queries (e.g., tokens 1066–1092) attend predominantly to nearby keys within a local window (approximately tokens 900–1092), exhibiting the characteristic local attention pattern. However, queries at tokens 1068 and 1090 deviate markedly from this pattern: they redirect attention toward globally distant keys (approximately tokens 100–450), corresponding to earlier segments of the reasoning trace. These tokens exemplify TRT, in which the model revisits previously formulated reasoning contexts to maintain global coherence.

To quantify this distinction, we measure the attention distance for each query, defined as the distance between the query position and the positions of its top-$K$ attended keys. Figure [1](#)(b) presents the distribution of attention distances for local and global queries across multiple attention heads. Local queries exhibit concentrated distance distributions centered near zero, reflecting attention focused on neighboring positions. In contrast, global queries attend to a substantially wider range of key positions, resulting in distributions that are clearly separated from those of local queries. This separation confirms that TRT represents a qualitatively distinct attention behavior that cannot be captured by meth-

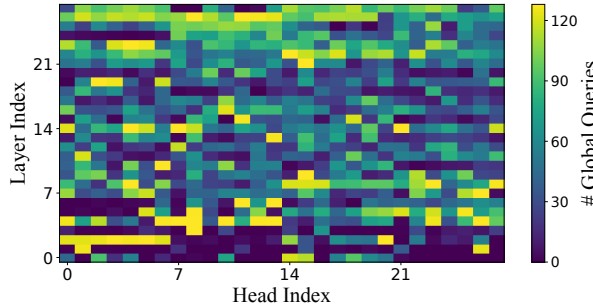

Figure 2. Output text tokens with Top-$K$ attention weights for each query at layer 18, head 16 (AIME24 sample-0 on R1-Distill-Qwen-7B). The full text is provided in Appendix E.

ods focusing solely on local context.

Figure 2 provides a concrete illustration of this phenomenon. For local queries (tokens 1089 and 1091), the top-$K$ attended tokens are concentrated on recent positions involved in ongoing calculations. For global queries corresponding to TRT (tokens 1068 and 1090), attention is redirected to earlier segments containing task-solving plans and problem constraints formulated at the beginning of the reasoning process. This re-attention to distant context is essential for maintaining coherence across extended reasoning traces.

Figure 3 shows the number of global queries over output token positions 512–639 for each layer and head on AIME24 sample-0 using R1-Distill-Qwen-7B. Global queries appear with varying frequencies across multiple layers and heads, rather than being concentrated in a single layer or attention head. This indicates that the global attention patterns associated with TRT are not an isolated behavior of a particular model component, but a recurring phenomenon that can emerge across different parts of the model throughout the reasoning trace.

**Implications for existing methods.** The existence of TRT reveals a fundamental limitation of prior KV cache compression methods. Approaches such as RPC (Song et al., 2025) and R-KV (Cai et al., 2025) rely on recent queries—queries collected from tokens immediately preceding the eviction step—to estimate which KV pairs will be important in subsequent decoding. While recent queries predominantly consist of local queries, they occasionally include global queries by chance. However, because global queries corresponding to

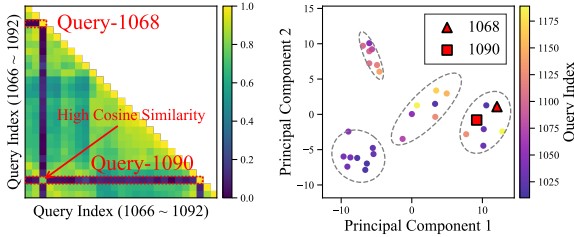

Figure 3. Occurrence distribution of global queries across layers and heads (AIME24 sample-0 on R1-Distill-Qwen-7B). Additional details are provided in Appendix B.

(a) Cosine similarity of queries    (b) PCA of queries

Figure 4. (a) Cosine similarity between queries within specific decoding step intervals of Layer-18 Head-16. (b) PCA analysis of queries with low average cosine similarity in Layer-18 Head-16.

TRT occur sporadically and unpredictably throughout the reasoning trace, recent-query-based methods systematically fail to anticipate which distant KV pairs will be revisited by future TRT. This leads to premature eviction of KV pairs that prove essential later, degrading reasoning quality under constrained memory budgets.

### 3.2. Geometric Structure of Global Queries

We now investigate whether global queries share structural properties that could enable their efficient representation. Specifically, we analyze the similarity structure of global queries in the embedding space.

Figure 4(a) shows the pairwise cosine similarity between queries within a decoding interval, computed using pre-RoPE query states to isolate geometric similarity from positional effects. While most queries exhibit high similarity to their immediate neighbors—reflecting the dominance of local queries—global queries, such as those at tokens 1068 and 1090, stand out by having substantially lower similarity to their surrounding queries. Crucially, despite their dissimilarity to their neighbors, these global queries exhibit high mutual similarity. This observation suggests that global queries form a coherent subset that is geometrically distinct from the majority of local queries.

To further characterize this structure, we project query states across decoding steps into a two-dimensional space using

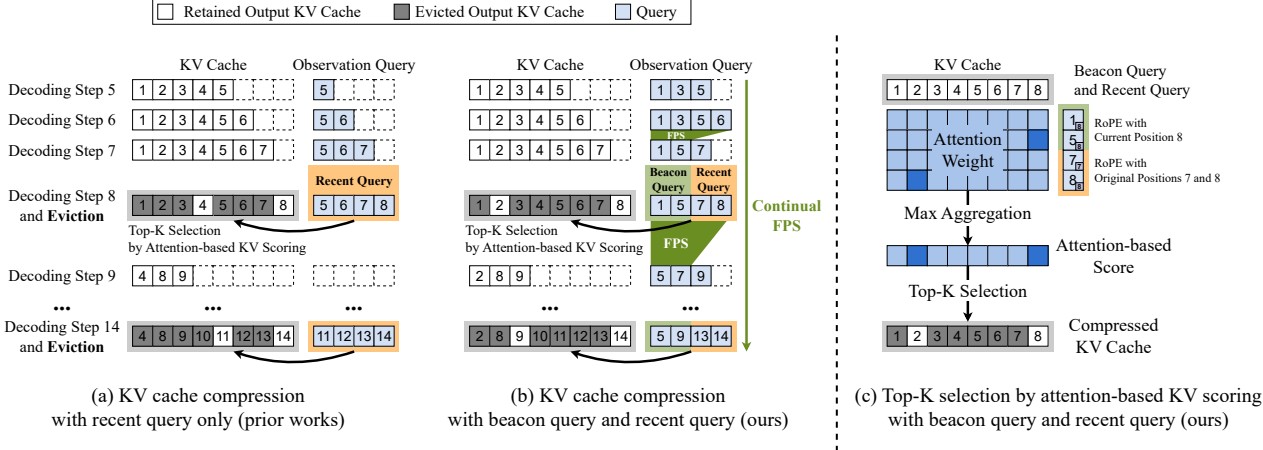

*Figure 5.* Illustration of KV cache compression methods for LRMs during long decoding. (a) Prior works (e.g., RPC) calculate attention-based scores for KV cache eviction relying solely on recent queries. (b) BeaconKV (Ours) computes the attention-based scores using beacon queries and recent queries. (c) Top-$K$ selection process by attention-based scores using beacon queries and recent queries.

Principal Component Analysis (PCA). Figure 4(b) visualizes the resulting projections, revealing that global queries cluster into a small number of similarity groups rather than being randomly scattered. Notably, queries at tokens 1068 and 1090—both corresponding to TRT—are located in close proximity within the same cluster, confirming that global queries with similar attention patterns share consistent geometric properties in the query embedding space.

**Beacon queries.** The observed clustering pattern motivates the notion of beacon queries. The diverse set of global queries that may arise throughout extended reasoning can be effectively represented by a compact set of beacon queries—representative queries for each global query cluster. By maintaining beacon queries alongside recent queries, a KV cache compression method can anticipate which KV pairs will be revisited by future global queries, even without storing the complete query history. The beacon queries serve as geometric landmarks in the query space, indicating which distant KV pairs should be retained to support TRT during subsequent decoding. This insight forms the foundation of our proposed method, BeaconKV.

## 4. Method

Building on these observations, we propose BeaconKV, a training-free KV cache compression framework that mitigates the memory bottleneck of LRMs by leveraging beacon queries. Our key insight is that reasoning-critical context is revisited by global queries that form clusters in the pre-RoPE query space. BeaconKV therefore augments the standard "recent query" baseline with beacon queries—a compact set of representative queries that anticipate future attention shifts. The detailed algorithm is provided in Appendix D.

### 4.1. Periodic KV Cache Eviction for LRMs

LRMs generate extended CoT to solve complex problems, resulting in a KV cache that grows linearly with decoding length. To operate within constrained GPU memory, the KV cache must be compressed periodically. Prior compression methods (Li et al., 2024; Song et al., 2025; Cai et al., 2025) typically perform eviction when the cache size reaches a budget $B_{\mathrm{KV}}^{\max}$. As illustrated in Figure 5(a), these methods construct an observation query set $Q_{\mathrm{obs}}$ utilizing only the most recent queries (e.g., the last 32 tokens). Consequently, they assign low importance scores to distant tokens that are not currently attended to, leading to the permanent loss of critical context. BeaconKV addresses this by expanding the observation window to include historical reference points, as shown in Figure 5(b), ensuring that "Thought Revisiting Tokens" are preserved even when they are temporally distant.

### 4.2. Observation Query Selection via Continual FPS

To capture the attention patterns of TRTs, BeaconKV constructs a more comprehensive observation query set containing two components: (1) recent queries ($Q_{\mathrm{recent}}^{\mathrm{pre}}$) to maintain local coherence, and (2) beacon queries ($Q_{\mathrm{beacon}}^{\mathrm{pre}}$) to represent the global query clusters identified in our geometric analysis.

**Empirical Motivation for FPS-based Selection.** BeaconKV leverages beacon queries selected from previously generated pre-RoPE queries for KV scoring. To identify a representative subset, we employ FPS based on cosine similarity across all pre-RoPE queries generated up to a given decoding step. Subsequently, we instantiate the observation query set $Q_{\mathrm{obs}}$ by integrating these FPS-selected queries with recent queries. Figure 6(a) reports the maxi-

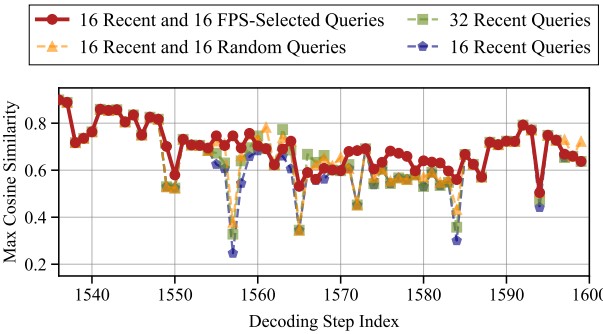

(a) Maximum cosine similarity between observation query set and query at the decoding step after eviction

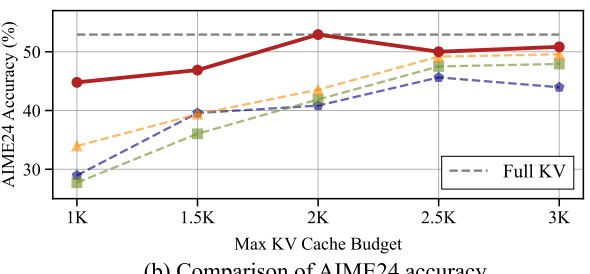

(b) Comparison of AIME24 accuracy across observation query set selections

*Figure 6.* (a) Maximum cosine similarity to future decoding step queries and (b) AIME24 accuracy on R1-Distill-Qwen-7B under different past query set selections.

mum cosine similarity between the observation query set and the query at each decoding step after eviction. Among several construction strategies, the configuration of 16 Recent + 16 FPS-Selected Queries exhibits high maximum cosine similarity for most decoding steps, indicating that its observation queries remain geometrically close to subsequent future queries. Figure 6(b) compares the accuracy under a constrained KV cache budget across these strategies. Consistent with the similarity analysis, the 16 Recent + 16 FPS strategy achieves the highest accuracy across all budgets. This suggests that effective KV scoring is driven more by query representativeness (geometric coverage) than by simply increasing recent query quantity.

**Efficient Implementation: Continual FPS.** While FPS is effective, running it over the entire history of accumulated queries (Naive FPS) is memory-intensive, particularly for LRMs using Grouped Query Attention (GQA), where query states are numerous. To mitigate this, we introduce Continual FPS, a memory-efficient online algorithm illustrated in Figure 5(b). Rather than storing all queries, each attention head maintains a small, bounded buffer. When this buffer reaches a maximum capacity $B_Q^{\max}$, we trigger an FPS step to downsample it back to a minimum size $B_Q^{\min}$, retaining only the most geometrically distinctive queries:

$$Q_{\text{obs}}^{\text{pre}} \leftarrow \text{FPS}\big(Q_{\text{obs}}^{\text{pre}}, B_Q^{\min}\big), \tag{9}$$

This "fill-and-compress" procedure ensures that $Q_{\text{beacon}}^{\text{pre}}$ con-

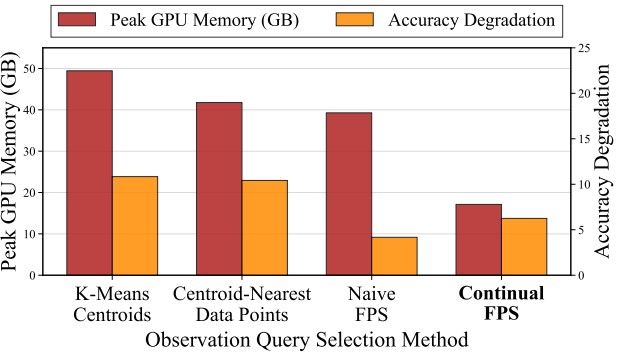

*Figure 7.* Peak GPU memory and accuracy degradation on R1-Distill-Qwen-7B under different observation query selection methods. We compare Continual FPS against K-Means Centroids, Centroid-Nearest Data Points, and Naive FPS.

tinuously evolves to represent the span of the reasoning trace without unbounded memory growth. We validate this design in Figure 7, which demonstrates that Continual FPS achieves accuracy comparable to ideal offline sampling (e.g., K-Means or Naive FPS) while reducing peak GPU memory usage by significantly minimizing the query history footprint.

### 4.3. Attention-Based KV Scoring with Beacon Queries

When the KV cache limit is reached at decoding step $t$, BeaconKV computes importance scores to determine which pairs to retain.

**Query Construction and Alignment.** We construct the observation query set $Q_{\text{obs}}$ by combining the accumulated beacon queries and the most recent queries. Crucially, we apply Rotary Positional Embeddings (RoPE) differentially to align these queries with the current reasoning state. Beacon queries are aligned to the current decoding step $t$. This allows us to simulate whether a future TRT (represented by the beacon) that occurs now would access the cached keys. Recent queries are kept at their original generation positions $\tau$. This preserves the standard local attention signals.

**Importance Scoring via Max-Aggregation.** Using $Q_{\text{obs}}$, we compute the attention weights $W \in \mathbb{R}^{|Q_{\text{obs}}| \times L}$ against the current KV cache. For models using GQA, where multiple query heads share a single KV head group $g$, we aggregate scores across all queries and heads in the group. Figure 5(c) illustrates our aggregation strategy. We employ max-pooling rather than mean-pooling across the observation queries. Since TRTs are sparse, event-driven by specific global queries, their attention signals are high-magnitude but rare. Mean-pooling would dilute these critical signals against the background noise of other queries. Max-pooling ensures that if any beacon query identifies a KV pair as important, that pair is preserved. Based on these scores, we retain the top-$K$ KV pairs alongside a small window of recent tokens (to ensure local fluency) and evict the rest.

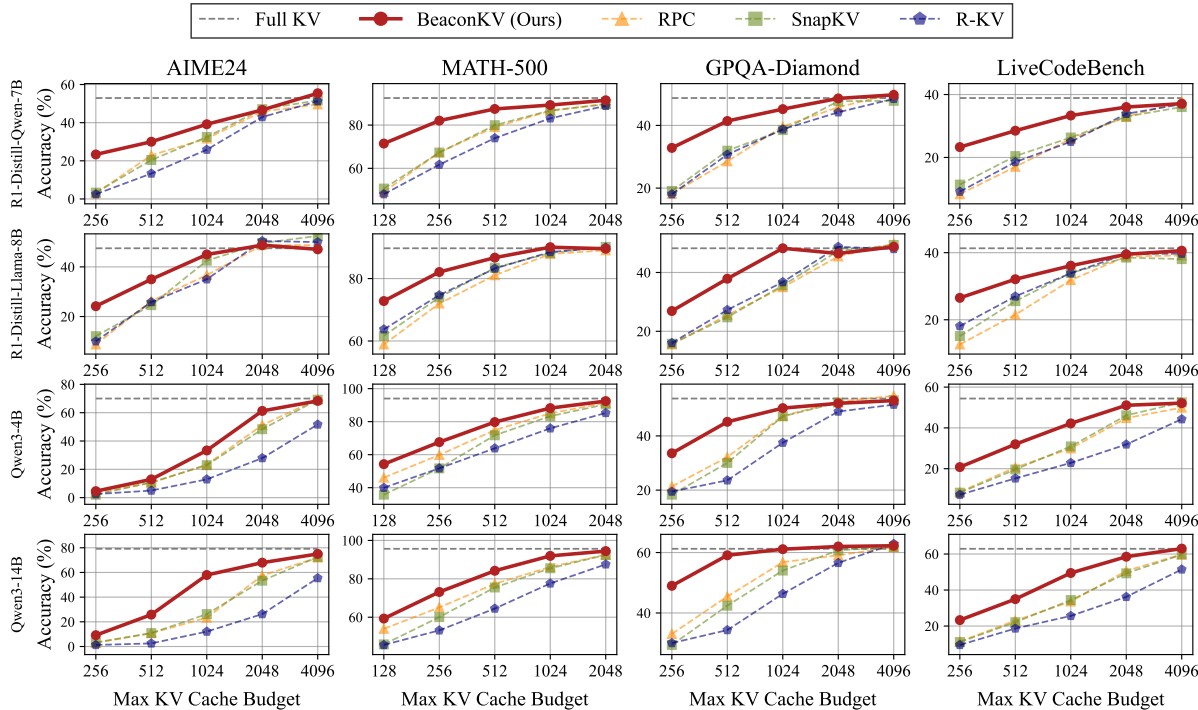

*Figure 8.* Accuracy comparison of BeaconKV against RPC, SnapKV, and R-KV. We evaluate the accuracy on R1-Distill-Qwen-7B, R1-Distill-Llama-8B, Qwen3-4B and Qwen3-14B across four reasoning tasks: AIME24, MATH-500, GPQA-Diamond, and LiveCodeBench.

This strategy allows BeaconKV to aggressively compress memory while preserving the "thought revisiting" pathways essential for deep reasoning.

## 5. Experiments

### 5.1. Experimental Setup

**Models and Datasets.** We evaluate BeaconKV on four open-source large reasoning models (LRMs): R1-Distill-Qwen-7B, R1-Distill-Llama-8B (Guo et al., 2025), Qwen3-4B, and Qwen3-14B (Yang et al., 2025). We consider various reasoning domains and use AIME24 (AIME, 2025) and MATH-500 (Hendrycks et al., 2021) for mathematics, LiveCodeBench (Jain et al., 2025) for coding, and GPQA-Diamond (Rein et al., 2024) for biology, physics, and chemistry. We report the average pass@1 accuracy over 8 runs for AIME24 and over 4 runs for the remaining benchmarks. We set the maximum number of generated tokens to 32,768 and use Top-$p$ sampling with $p = 0.95$ and temperature 0.6.

**Budget Configuration.** We manage KV cache compression with a maximum cache budget $B_{\mathrm{KV}}^{\max}$ and set the minimum budget to $B_{\mathrm{KV}}^{\min} = \frac{7}{8} B_{\mathrm{KV}}^{\max}$. When the cache size of the output tokens reaches $B_{\mathrm{KV}}^{\max}$, we evict the KV cache corresponding to the $\frac{1}{8} B_{\mathrm{KV}}^{\max}$ output tokens.

**Baselines.** We evaluate the accuracy of BeaconKV against several competitive KV cache compression meth-

ods. SnapKV and RPC perform KV cache eviction using attention-based scores, while R-KV employs a joint eviction strategy that integrates per-token redundancy scores.

**Implementation Details.** For BeaconKV, we set the maximum beacon query token budget to 32 and the minimum to 16. The recent query token budget is fixed at 16. When KV cache compression is enabled, R-KV retains the KV cache for the most recent 8 tokens, following the configuration used in the original R-KV paper. For all other methods, we retain the most recent 32 tokens.

### 5.2. Performance on Reasoning Tasks

Figure 8 reports accuracy under varying KV cache budgets. Across most benchmarks and models, BeaconKV achieves the highest accuracy at the same budget, with the largest gains in the low-budget regime, substantially outperforming reasoning-oriented methods such as RPC and R-KV. The largest accuracy gain over existing compression methods reaches 31.7 percentage points, observed on Qwen3-14B for AIME24 with a maximum KV cache budget of 1024. This advantage arises from BeaconKV's beacon queries—geometrically distinctive past queries selected via Continual FPS—that capture global attention patterns associated with Thought Revisiting Tokens (TRT). In contrast, recent-query–based methods rely on locally concentrated queries and thus tend to evict distant yet reusable reasoning trajectories, whereas BeaconKV retains such KV pairs to

| Method | $(n_{recent}, n_{beacon})$ | Accuracy (%) ↑ | Latency (s) ↓ |
|---|---|---|---|
| | (1, 31) | 63.5 | 10871.3 |
| | (4, 28) | 66.5 | 5624.5 |
| | (8, 24) | 64.8 | 4762.1 |
| BeaconKV | **(16, 16)** | **64.6** | **4355.7** |
| | (24, 8) | 62.3 | 4194.5 |
| | (28, 4) | 58.1 | 4179.5 |
| | (31, 1) | 55.6 | 4192.3 |
| RPC | (32, 0) | 51.3 | 4127.2 |

*Table 1.* Ablation study on the number of beacon queries ($n_{beacon}$) and recent queries ($n_{recent}$) on Qwen3-4B for AIME24, with the maximum KV cache budget set to 2048. We report both accuracy and latency to identify the optimal configuration. The setting ($n_{recent} = 16, n_{beacon} = 16$) achieves the best trade-off.

| Aggregation | Max KV Cache Budget | | | | |
|---|---|---|---|---|---|
| | 4096 | 2048 | 1024 | 512 | 256 |
| Max | **55.4** | 46.7 | **39.2** | **30.0** | **23.3** |
| Mean | 52.5 | **50.8** | 36.7 | 27.5 | 18.8 |

*Table 2.* Ablation study on aggregation in BeaconKV under different maximum KV cache budgets on R1-Distill-Qwen-7B for AIME24. We compare using Max vs. Mean aggregation consistently across heads and queries.

maintain robust reasoning accuracy under tight budgets.

## 5.3. Ablation Study

We first study the trade-off between beacon queries ($n_{beacon}$) and recent queries ($n_{recent}$) for observation query selection. As shown in Table 1, over-allocating queries to beacons reduces the capacity to capture short-term context, resulting in both higher latency and lower accuracy (e.g., (1, 31) yields 63.5% accuracy with 10871.3s latency). In contrast, a balanced configuration ($n_{recent}, n_{beacon}) = (16, 16)$ achieves the favorable trade-off, attaining 64.6% accuracy with substantially lower latency (4355.7s), highlighting the importance of jointly preserving global and local query signals.

We further ablate aggregation strategies for KV importance scoring under different KV cache budgets. Table 2 shows that Max aggregation generally outperforms Mean aggregation, with larger gaps observed in most low-budget settings (e.g., 23.3 for Max aggregation compared to 18.8 for Mean aggregation at a 256-token budget). This behavior arises because important KV pairs are often emphasized by individual beacon queries; Mean aggregation dilutes such sparse but high-value signals, whereas Max aggregation preserves them, ensuring that the most salient features are retained.

To examine whether BeaconKV's gains arise from dynamically capturing evolving global query patterns, we compare it with Initial+Recent, a query-based scoring baseline that preserves both initial and recent queries. Unlike standard recent-query-based eviction methods that estimate KV importance only from the most recent queries, Initial+Recent

| Model | Method | AIME 24 | MATH -500 | GPQA -Diamond | LiveCode Bench |
|---|---|---|---|---|---|
| R1-Distill-7B | BeaconKV | 39.17 | 87.45 | 45.20 | 33.33 |
| | Initial+Recent | 20.42 | 70.60 | 42.80 | 28.95 |
| R1-Distill-8B | BeaconKV | 45.00 | 86.70 | 48.23 | 36.11 |
| | Initial+Recent | 44.58 | 81.40 | 45.07 | 33.16 |
| Qwen3-4B | BeaconKV | 33.33 | 79.75 | 50.25 | 42.21 |
| | Initial+Recent | 19.58 | 55.95 | 49.87 | 33.93 |
| Qwen3-14B | BeaconKV | 57.92 | 84.20 | 61.11 | 49.46 |
| | Initial+Recent | 23.75 | 67.30 | 60.23 | 40.23 |

*Table 3.* Accuracy comparison between BeaconKV and Initial+Recent across models and reasoning tasks under a maximum KV cache budget of 1024.

| Method | Max KV Budget | Batch Size | Throughput (tokens/s) | Decode Latency | Peak Mem. | Acc. |
|---|---|---|---|---|---|---|
| Full KV | – | 14 | 82.3 | 5573.4 | 77.0 | 54.4 |
| BeaconKV | 2K | 14 | 356.4 | 1287.3 | 13.3 | 51.1 |
| RPC vs. BeaconKV | | | | | | |
| RPC | 2K | 192 | 725.4 | 8672.5 | 79.0 | 44.8 |
| BeaconKV | 2K | 192 | 704.8 | 8926.9 | 79.3 | 51.1 |
| RPC | 1K | 320 | 1380.8 | 7593.7 | 72.0 | 29.9 |
| BeaconKV | 1K | 320 | 1345.9 | 7790.9 | 72.5 | 42.2 |

*Table 4.* Efficiency evaluation on Qwen3-4B with a generation length of 32K. We compare the throughput, decoding latency (s), peak GPU memory usage (GB), and LiveCodeBench accuracy of Full KV, RPC, and BeaconKV.

builds the observation query set from both initially generated queries and recent queries. The resulting attention weights are then used to score KV pairs and determine which entries should be retained.

As shown in Table 3, BeaconKV consistently outperforms Initial+Recent across the evaluated models and reasoning tasks. This indicates that simply preserving queries from the beginning of context is not sufficient to explain BeaconKV's gains. Although initial queries provide access to early reasoning context, they form a fixed and limited set of historical references and cannot capture the diverse global revisiting patterns that emerge during long-horizon reasoning. In contrast, BeaconKV continually selects geometrically diverse beacon queries from the evolving query history, allowing it to identify KV pairs that are likely to be revisited by future Thought Revisiting Tokens. These results suggest that the gains from Continual FPS mainly stem from its ability to dynamically capture emerging global query patterns, rather than from merely preserving the beginning of the reasoning trajectory.

## 5.4. Efficiency Evaluation

We evaluate system efficiency by comparing BeaconKV with Full KV and RPC on Qwen3-4B with a generation length of 32K tokens using a single NVIDIA A100 80GB GPU. Table 4 reports throughput, decoding latency, peak

GPU memory, and LiveCodeBench accuracy across different batch sizes and KV cache budgets. Full KV exhibits a severe memory bottleneck under long decoding, reaching 77.0 GB peak memory at batch size 14 and failing to scale further. In contrast, BeaconKV with a 2K KV budget reduces peak memory from 77.0 GB to 13.3 GB ($5.8\times$), leading to higher throughput ($82.3 \rightarrow 356.4$ tokens/s) and lower decoding latency ($5573.4 \rightarrow 1287.3$s), and enabling larger batch sizes.

Under matched KV budgets, BeaconKV achieves system efficiency comparable to RPC while delivering higher accuracy. At a 2K budget (batch size 192), BeaconKV matches RPC in throughput and memory usage, while improving LiveCodeBench accuracy by +6.3 points. At a tighter 1K budget (batch size 320), BeaconKV again achieves similar throughput and memory, but yields a larger accuracy gain of +12.3 points. These results demonstrate that BeaconKV preserves system-level efficiency while substantially improving generation quality under constrained KV cache budgets.

## 6. Related Work

**KV Cache Compression.**  To mitigate the memory overhead of the KV cache in long-context inference, prior KV cache compression methods evict the cache using attention-based importance scores (Zhang et al., 2023; Oren et al., 2024; Li et al., 2024; Kim et al., 2024; 2026b;c; Yan et al., 2026).  More recent work targets large reasoning models (LRMs), where long generations with extended Chain-of-Thought rapidly expand the KV cache.  For instance, RPC (Song et al., 2025) applies attention-based scoring for KV compression in LRMs. Beyond attention-based scores, several approaches incorporate redundancy or recurrence signals. KeyDiff (Park et al., 2026) proposes a compression strategy driven by key similarity. R-KV (Cai et al., 2025) combines an attention-based score with a redundancy score through a joint selection rule. LazyEviction (Zhang et al., 2025a) performs lagged eviction by leveraging recurring importance patterns.

Another line of work uses a lightweight trainable module to learn policies for KV cache eviction. TRIM-KV (Bui et al., 2026), LightThinker (Zhang et al., 2025b), and Fast KVzip (Kim et al., 2026a) each introduce small gating or compression modules—trained via distillation or supervised fine-tuning on top of frozen backbone weights—to predict token-level importance or compress reasoning context during long reasoning with extended Chain-of-Thought. However, these methods require task-specific training and may exhibit limited generalization across diverse reasoning domains. In contrast, BeaconKV requires no training, instead exploiting the intrinsic geometric structure of query embeddings to identify critical KV pairs without any task-specific optimization.

**Sparse Attention for Large Reasoning Models.**  For large reasoning models (LRMs), sparse attention methods reduce attention computation by restricting each query to attend only to a subset of the KV cache of the reasoning process (Tang et al., 2024; Jiang et al., 2024), while retaining the full KV cache in memory. Multipole Attention (Hooper et al., 2026) clusters the KV cache, attends exactly to selected KV entries, and approximates the rest. ReSA (Sun et al., 2025) mitigates accumulated errors via periodic dense rectification. SeerAttention-R (Gao et al., 2026) learns sparsity patterns using a self-distilled gating module.

**Adaptive Control of Reasoning Length.**  Orthogonal to KV cache compression, adaptive reasoning control strategies optimize the reasoning path by analyzing the characteristics of the Chain-of-Thought. DEER (Yang et al., 2026) proposes a training-free dynamic early exit mechanism at transition points, while SEAL (Chen et al., 2025) calibrates reasoning paths via steerable interventions. InftyThink (Yan et al., 2026) employs a training-based framework to enable deeper reasoning by interleaving short reasoning steps with intermediate summarization.

**Memory-Augmented Architectures.**  While the KV cache compression and sparse attention methods discussed above primarily optimize inference within the standard decoder-only transformer architecture, another paradigm expands long-context capability by introducing memory modules into the model architecture. Memory-augmented architectures such as Titans (Behrouz et al., 2026) and GNM (Bennett et al., 2026) use such memory modules as context storage, enabling the model to dynamically compress, store, and reuse important context. In contrast, BeaconKV does not introduce additional memory modules or modify the model architecture. Instead, it provides a training-free framework in which beacon queries, identified from the intrinsic geometry of the query space, serve as selectors that distinguish contextually important KV cache entries for retention.

## 7. Conclusion

We introduce BeaconKV, a training-free KV cache compression approach that alleviates memory bottlenecks in Large Reasoning Models (LRMs). Motivated by the observation of Thought Revisiting Tokens—where models re-attend to prior context to maintain reasoning coherence—BeaconKV leverages geometrically distinctive past queries to preserve critical KV cache. Across models and benchmarks, BeaconKV generally outperforms existing compression methods, reducing memory by up to $5.8\times$ while achieving accuracy close to full KV inference and improving throughput by over $4.3\times$.

## Acknowledgements

This work was supported by the National Research Foundation of Korea (NRF) grant funded by the Korea government (MSIT) (No. RS-2025-00561961 and No. RS-2023-00260527). This work was also supported by the Institute of Information & Communications Technology Planning & Evaluation (IITP) grant funded by the Korea government (MSIT) (under the Artificial Intelligence Semiconductor Support Program to Nurture the Best Talents (IITP-2026-RS-2023-00253914), and No. RS-2025-02214497, Development of low-level optimization program API technology for AI semiconductors, and No. RS-2019-II190421, AI Graduate School Support Program (Sungkyunkwan University)). This research was also supported by the Advanced GPU Utilization Support Program funded by the Government of the Republic of Korea (Ministry of Science and ICT).

## Impact Statement

This work advances the field of efficient Large Reasoning Model (LRM) inference by addressing the critical memory bottlenecks associated with extended Chain-of-Thought generation. By reducing peak GPU memory usage and significantly enhancing throughput, BeaconKV lowers the hardware barrier for deploying advanced reasoning models, thereby broadening access to capabilities that previously required high-end GPU clusters. In addition, our training-free compression method contributes to sustainable AI development by improving the energy efficiency of long-horizon reasoning, helping to mitigate the environmental footprint associated with the large-scale deployment of foundation models. At the same time, making long-horizon reasoning more cost-efficient may also lower the cost of scaling unintended or undesirable uses. Furthermore, since our evaluation is conducted primarily on open-source LRMs and benchmark settings, further assessment is needed to understand how the benefits of BeaconKV transfer to broader generation workloads.

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

# A. Additional Experimental Results

## A.1. Comparison with SnapKV

We provide an additional efficiency comparison with SnapKV (Li et al., 2024), a representative attention-based KV cache eviction baseline. All methods are evaluated on Qwen3-4B with a maximum KV cache budget of 1K and a batch size of 320. As shown in Table 5, BeaconKV achieves substantially higher LiveCodeBench accuracy than both SnapKV and RPC while maintaining comparable system efficiency. Specifically, BeaconKV improves accuracy from 30.9% with SnapKV and 29.9% with RPC to 42.2%, while exhibiting similar throughput and peak GPU memory usage. Although BeaconKV incurs a modest increase in decoding latency due to beacon-query-based scoring, this overhead is small relative to the accuracy improvement, demonstrating that BeaconKV provides a more favorable accuracy–efficiency trade-off under the same memory budget.

| Method | Max KV Budget | Batch Size | Throughput (tokens/s) | Decode Latency | Peak Mem. | LiveCodeBench Acc. |
|---|---|---|---|---|---|---|
| SnapKV | 1K | 320 | 1380.7 | 7594.5 | 72.4 | 30.9 |
| RPC | 1K | 320 | 1380.8 | 7593.7 | 72.0 | 29.9 |
| BeaconKV | 1K | 320 | 1345.9 | 7790.9 | 72.5 | 42.2 |

*Table 5.* Efficiency and accuracy comparison among SnapKV, RPC, and BeaconKV on Qwen3-4B under a 1K KV cache budget.

## A.2. Output Length Statistics

Table 6 reports the average number of generated output tokens for each model and task. The results show that reasoning benchmarks often induce long Chain-of-Thought generations, frequently reaching several thousand tokens and becoming especially long on AIME24 and LiveCodeBench. Since the KV cache grows linearly with the decoding length, these output length statistics illustrate the practical memory pressure imposed by LRM inference. This further motivates BeaconKV, which reduces the KV cache footprint while preserving reasoning-relevant context during long-horizon generation.

| Model | AIME24 | MATH-500 | GPQA-Diamond | LiveCodeBench |
|---|---|---|---|---|
| R1-Distill-Qwen-7B | 13459 | 4069 | 8210 | 11720 |
| R1-Distill-Llama-8B | 14253 | 4241 | 8755 | 11983 |
| Qwen3-4B | 14675 | 5342 | 6516 | 14100 |
| Qwen3-14B | 13801 | 4799 | 5358 | 12623 |

*Table 6.* Average Number of Output Tokens by Model and Task.

# B. Additional Details on the Occurrence Distribution of Global Queries

To provide an additional statistical analysis of Thought Revisiting Tokens (TRT), we measure the occurrence frequency of global queries across layers and attention heads. For each target query, we compute its attention weights over the keys available up to that decoding step and identify the top-$K$ most attended positions among output tokens, where $K = 150$. We then measure the token distance between the query position and each of these top-$K$ positions, and take the mean of these distances. A query is classified as local if the resulting mean attention distance is at most 200, and as global otherwise.

# C. Limitations and Future Work

Our evaluation primarily focuses on long-horizon reasoning tasks using open-source Large Reasoning Models. Therefore, the effectiveness of BeaconKV on standard non-reasoning language modeling tasks, such as long-context retrieval, text summarization, and general long-context generation, remains insufficiently explored. Further experiments are needed to determine whether the performance gains achieved by BeaconKV generalize to broader workloads. In addition, BeaconKV involves several hyperparameters, including the number of beacon queries, the number of recent queries, and the KV cache budget. While our main experiments use fixed configurations, the optimal settings may vary across models and tasks. A more systematic analysis of hyperparameter sensitivity is left for future work. Additionally, it would be promising to develop adaptive compression strategies that dynamically adjust these hyperparameters during inference, such as varying the number of beacon queries or the KV cache budget at each compression rather than keeping them fixed throughout generation.

# D. Algorithms

### D.1. Farthest Point Sampling (FPS)

We use Farthest Point Sampling (FPS) to retain a compact set of diverse representative queries from the accumulated pre-RoPE query set $Q = \{q_1, \ldots, q_N\}$. Given a target query budget $m$, FPS greedily selects queries that are least similar to the current selected set under cosine similarity. The algorithm first computes the average cosine similarity of each query, $\text{avg}_i = \frac{1}{N} \sum_{j=1}^{N} \cos(q_i, q_j)$, and initializes the compressed set with $q_p$, where $p = \arg\min_i \text{avg}_i$. This choice favors a non-redundant query that is least aligned with the overall query set. FPS then maintains, for each unselected query $q_j$, its nearest-representative similarity $S_j = \max_{k \in \mathcal{I}_{comp}} \cos(q_j, q_k)$, where $\mathcal{I}_{comp}$ is the set of selected indices. At each iteration, it selects $r = \arg\min_{j \notin \mathcal{I}_{comp}} S_j$, i.e., the query farthest from the current compressed set, adds $q_r$ to $Q_{comp}$, and updates $S_j \leftarrow \max(S_j, \cos(q_j, q_r))$ for the remaining queries. By operating in the pre-RoPE query space, FPS captures geometric diversity among queries before positional rotations are applied. The resulting set $Q_{comp}$ provides a compact approximation of the historical query distribution while preserving representative query directions that may be important for subsequent KV cache scoring.

### D.2. BeaconKV

Algorithm 2 summarizes BeaconKV within a single GQA head group. The method uses two types of observation queries: beacon queries, which compactly represent historical query patterns, and recent queries, which preserve short-range decoding information. During prefill, all KV pairs are inserted into the cache, and each head initializes its observation set by applying FPS to the pre-RoPE query sequence:

$$Q_{\text{obs}}^{(h),\text{pre}} = \text{FPS}\big(\{q_{h,\tau}^{\text{pre}}\}_{\tau=1}^{T}, B_Q^{\min}\big), \quad B_Q^{\min} = n_{\text{beacon}}.$$

BeaconKV stores queries before RoPE so that their positional encoding can be assigned later depending on whether they are used as beacon or recent queries. During decoding, newly generated pre-RoPE queries are appended to the observation set until its size reaches

$$B_Q^{\max} = n_{\text{beacon}} + n_{\text{recent}}.$$

The set is then compressed back to $B_Q^{\min}$ representatives using FPS. When the KV cache reaches the boundary $B_{\text{KV}}^{\max} - n_{\text{recent}}$, these representatives are fixed as beacon queries, and the following $n_{\text{recent}}$ queries are stored as recent queries. Once the KV cache reaches the maximum budget $B_{\text{KV}}^{\max}$, BeaconKV forms the group-level observation set as

$$Q_{\text{obs}} = \{\text{RoPE}(q, t) \mid q \in Q_{\text{beacon}}^{g,\text{pre}}\} \cup \{\text{RoPE}(q, \tau) \mid \langle q, \tau \rangle \in Q_{\text{recent}}^{g,\text{pre}}\}.$$

Beacon queries are rotated at the current step $t$ to act as global selectors, while recent queries keep their original positions $\tau$. The importance of each cached KV position $j$ is then computed by max-pooling attention weights over heads and observation queries:

$$\text{Score}[j] = \max_{h \in g} \max_{q \in Q_{\text{obs}}} W[h, q, j].$$

BeaconKV always preserves prefix tokens and the most recent tokens,

$$\mathcal{I}_{\text{keep}} = \{1, \ldots, n_{\text{prefix}}\} \cup \{L - n_{\text{recent}} + 1, \ldots, L\},$$

and fills the remaining budget with the top-scoring KV positions. After eviction, the retained beacon and recent queries are merged and compressed again with FPS, allowing BeaconKV to continually maintain a compact set of query representatives throughout long-horizon decoding.

# E. Token-Level Visualization of Thought Revisiting Tokens

Figure 9 shows the full token-level visualization corresponding to the example in Figure 2. We highlight the top-$K$ attended tokens for representative global queries (tokens 1068 and 1090) and local queries (tokens 1089 and 1091) in an AIME24 reasoning trace. Global queries revisit distant spans containing the original problem constraints and high-level solving plan, whereas local queries mainly focus on nearby steps.

---

**Algorithm 1** Farthest Point Sampling (FPS)

---

**Input:** pre-rope query set $Q = \{q_1, \ldots, q_N\}$, Min Query Budget $m$
**Output:** compressed query set $Q_{comp}$

// Select initial query with the smallest average cosine similarity
**for** $i = 1$ **to** $N$ **do**
  $\text{avg}_i \leftarrow \frac{1}{N} \sum_{j=1}^{N} \cos(q_i, q_j)$                         // cosine similarity denoted as $\cos(\cdot, \cdot)$
**end for**
$p \leftarrow \arg\min_i \text{avg}_i$
$Q_{comp} \leftarrow \{q_p\}; \quad \mathcal{I}_{comp} \leftarrow \{p\}$

// Nearest-similarity initialization
**for** $j = 1$ **to** $N$ **do**
  $S_j \leftarrow \cos(q_j, q_p)$                         // similarity to the nearest selected (currently $q_p$)
**end for**

// Greedy farthest expansion under cosine similarity
**while** $|\mathcal{I}_{comp}| < m$ **do**
  $r \leftarrow \arg\min_{j \notin \mathcal{I}_{comp}} S_j$                         // farthest = least similar
  $Q_{comp} \leftarrow Q_{comp} \cup \{q_r\}$
  $\mathcal{I}_{comp} \leftarrow \mathcal{I}_{comp} \cup \{r\}$
  **for** $j \notin \mathcal{I}_{comp}$ **do**
    $S_j \leftarrow \max\left(S_j, \cos(q_j, q_r)\right)$                         // update nearest similarity
  **end for**
**end while**
**return** $Q_{comp}$

---

---

**Algorithm 2** BeaconKV within a GQA Head Group

---

**Input:** head group $g$, step $t$, pre-RoPE queries $Q^{\text{pre}}$, shared KV cache $C_{\text{KV}}$, $\{Q_{\text{obs}}^{(h),\text{pre}}\}_{h \in g}$, $\{Q_{\text{beacon}}^{(h),\text{pre}}\}_{h \in g}$, $\{Q_{\text{recent}}^{(h),\text{pre}}\}_{h \in g}$ (tuples $\langle q^{\text{pre}}, \tau \rangle$), $n_{\text{prefix}}, n_{\text{beacon}}, n_{\text{recent}}, B_{\text{KV}}^{\max}, B_{\text{KV}}^{\min}$

**Output:** updated $C_{\text{KV}}, \{Q_{\text{obs}}^{(h),\text{pre}}, Q_{\text{beacon}}^{(h),\text{pre}}, Q_{\text{recent}}^{(h),\text{pre}}\}_{h \in g}$

$B_Q^{\max} \leftarrow n_{\text{beacon}} + n_{\text{recent}}, \quad B_Q^{\min} \leftarrow n_{\text{beacon}}$

**if** $|Q^{\text{pre}}| \neq 1$ **then**
    $C_{\text{KV}} \leftarrow C_{\text{KV}} \cup \{(K_\tau, V_\tau) \mid \tau = 1, \ldots, T\}$
    **for each** $h \in g$ **do**
        $Q_{\text{obs}}^{(h),\text{pre}} \leftarrow \text{FPS}(\{q_{h,\tau}^{\text{pre}}\}_{\tau=1}^T, B_Q^{\min})$
        $Q_{\text{beacon}}^{(h),\text{pre}} \leftarrow \emptyset, \quad Q_{\text{recent}}^{(h),\text{pre}} \leftarrow \emptyset$
    **end for**
    **return** $C_{\text{KV}}, \{Q_{\text{obs}}^{(h),\text{pre}}, Q_{\text{beacon}}^{(h),\text{pre}}, Q_{\text{recent}}^{(h),\text{pre}}\}_{h \in g}$
**end if**

$C_{\text{KV}} \leftarrow C_{\text{KV}} \cup \{(K_t, V_t)\}, \quad L \leftarrow |C_{\text{KV}}|$
$\{q_h^{\text{pre}}\}_{h \in g} \leftarrow \{q_{h,1}^{\text{pre}}\}_{h \in g}$

**if** $L \leq B_{\text{KV}}^{\max} - n_{\text{recent}}$ **then**
    **for each** $h \in g$ **do**
        $Q_{\text{obs}}^{(h),\text{pre}} \leftarrow Q_{\text{obs}}^{(h),\text{pre}} \cup \{q_h^{\text{pre}}\}$
        **if** $|Q_{\text{obs}}^{(h),\text{pre}}| \geq B_Q^{\max}$ **then**
            $Q_{\text{obs}}^{(h),\text{pre}} \leftarrow \text{FPS}(Q_{\text{obs}}^{(h),\text{pre}}, B_Q^{\min})$
        **end if**
    **end for**
**end if**
**if** $L = B_{\text{KV}}^{\max} - n_{\text{recent}}$ **then**
    **for each** $h \in g$ **do**
        $Q_{\text{beacon}}^{(h),\text{pre}} \leftarrow \text{FPS}(Q_{\text{obs}}^{(h),\text{pre}}, B_Q^{\min})$
        $Q_{\text{recent}}^{(h),\text{pre}} \leftarrow \emptyset$
    **end for**
**end if**
**if** $B_{\text{KV}}^{\max} - n_{\text{recent}} < L \leq B_{\text{KV}}^{\max}$ **then**
    **for each** $h \in g$ **do**
        $Q_{\text{recent}}^{(h),\text{pre}} \leftarrow Q_{\text{recent}}^{(h),\text{pre}} \cup \{\langle q_h^{\text{pre}}, t \rangle\}$
    **end for**
**end if**

**if** $L = B_{\text{KV}}^{\max}$ **then**
    $Q_{\text{beacon}}^{g,\text{pre}} \leftarrow \bigcup_{h \in g} Q_{\text{beacon}}^{(h),\text{pre}}$
    $Q_{\text{recent}}^{g,\text{pre}} \leftarrow \bigcup_{h \in g} Q_{\text{recent}}^{(h),\text{pre}}$
    $Q_{\text{obs}} \leftarrow \{\text{RoPE}(q, t) \mid q \in Q_{\text{beacon}}^{g,\text{pre}}\} \cup \{\text{RoPE}(q, \tau) \mid \langle q, \tau \rangle \in Q_{\text{recent}}^{g,\text{pre}}\}$
    $W \leftarrow \text{AttnWeight}(Q_{\text{obs}}, C_{\text{KV}})$
    $\text{Score}[j] \leftarrow \max_{h \in g} \max_{q \in Q_{\text{obs}}} W[h, q, j] \quad, \forall j$
    $\mathcal{I}_{\text{keep}} \leftarrow \{1, \ldots, n_{\text{prefix}}\} \cup \{L - n_{\text{recent}} + 1, \ldots, L\}$
    $k_{\text{sel}} \leftarrow B_{\text{KV}}^{\min} - |\mathcal{I}_{\text{keep}}|$
    $\mathcal{I}_{\text{top}} \leftarrow \text{TopK}(\text{Score}, k_{\text{sel}}, \text{exclude} = \mathcal{I}_{\text{keep}})$
    $\mathcal{I}_{\text{final}} \leftarrow \text{Sort}(\mathcal{I}_{\text{keep}} \cup \mathcal{I}_{\text{top}})$
    $C_{\text{KV}} \leftarrow \text{Gather}(C_{\text{KV}}, \mathcal{I}_{\text{final}})$
    **for each** $h \in g$ **do**
        $Q_{\text{obs}}^{(h),\text{pre}} \leftarrow Q_{\text{beacon}}^{(h),\text{pre}} \cup \{q \mid \langle q, \tau \rangle \in Q_{\text{recent}}^{(h),\text{pre}}\}$
        $Q_{\text{obs}}^{(h),\text{pre}} \leftarrow \text{FPS}(Q_{\text{obs}}^{(h),\text{pre}}, B_Q^{\min})$
        $Q_{\text{beacon}}^{(h),\text{pre}} \leftarrow \emptyset, \quad Q_{\text{recent}}^{(h),\text{pre}} \leftarrow \emptyset$
    **end for**
**end if**
**return** $C_{\text{KV}}, \{Q_{\text{obs}}^{(h),\text{pre}}, Q_{\text{beacon}}^{(h),\text{pre}}, Q_{\text{recent}}^{(h),\text{pre}}\}_{h \in g}$

---

*Figure 9.* Full token-level visualization of top-$K$ attended positions for representative global and local queries in an AIME24 sample-0 on R1-Distill-Qwen-7B (Layer 18, Head 16). Colored underlines indicate the tokens most strongly attended by each query. Global queries corresponding to Thought Revisiting Tokens (tokens 1068 and 1090) revisit earlier problem constraints and high-level solving plans, whereas local queries (tokens 1089 and 1091) mainly attend to nearby steps.

