# OpenReview forum: "BeaconKV: Key-Value Cache Compression Guided by Beacon Queries for Efficient Large Reasoning Model Inference"
_ICML.cc/2026/Conference — ICML 2026 regular_

### Official Review · Reviewer_JcCz · 2026-03-09

**Soundness:** 3
**Presentation:** 2
**Significance:** 3
**Originality:** 2
**Overall Recommendation:** 4
**Confidence:** 4

**Summary:**

This paper proposes a training-free KV cache compression method named "BeaconKV," designed for large reasoning models with long Chain-of-Thought (CoT). The authors observe the phenomenon that the model will re-attend to distant historical tokens during the reasoning process, and also observe that the query vectors triggering TRT exhibit clustering characteristics in the embedding space. Based on these findings, the paper proposes using Continual Farthest Point Sampling (Continual FPS) to filter "Beacon Queries" from historical queries.

**Compliance With Llm Reviewing Policy:**

Affirmed.

**Final Justification:**

I decide to raise my score. The authors' experimental data resolved my main concerns. 1.The newly added Table R1 proves the necessity of this mechanism compared to the static baseline, while Table R2 demonstrates an excellent accuracy-latency trade-off. 2.Although the parameter robustness tests are still not comprehensive enough, the authors promised to explicitly state these application boundaries and limitations in the final version.

**Key Questions For Authors:**

1.Add experiments to prove that the overall system efficiency can still be improved after introducing the distance matrix computation.

2.Provide relevant experiments, such as only retaining Recent Tokens + the initial N tokens of the sequence, to prove that the gains brought by Continual FPS indeed originate from its dynamic capture capability, rather than merely from the setup of preserving the beginning.

3.Supplement the time-consumption ratio of the Continual FPS module under different sequence lengths (e.g., 32K, 64K, 128K).

4.It is recommended to supplement general NLP task datasets on standard long-text benchmarks to prove the effectiveness of the method.

**Limitations:**

yes

**Strengths And Weaknesses:**

Strengths of paper:

1.The identification of "thought revisiting tokens" highly aligns with current intuitions regarding CoT reasoning mechanisms (i.e., the model needs to backtrack to premises/plans).

2.The performance on benchmark datasets is impressive. In heavy reasoning tasks with high compression rates (e.g., AIME24), this method demonstrates strong context retention capabilities, outperforming the selected baseline models.

3.The training-free method requires no auxiliary models or fine-tuning, demonstrating good engineering practicality.

Weaknesses of paper:

1.The paper proposes that the model primarily "looks back" at the beginning part of the sequence. Why is it necessary to design a complex Continual FPS mechanism? A simple and easy heuristic method (e.g., a fixed retention of X%) could achieve the same effect.

2.The BeaconKV model needs to compute a distance matrix, introducing additional computational overhead. Will this overhead impact the model's inference latency? Calculating attention scores for Beacon Queries requires an additional matrix multiplication at each eviction step; there is a lack of quantitative experiments for this overhead.

3.In Table 3, BeaconKV only compares against RPC and Full KV. How does its performance compare to methods like SnapKV, H2O, and CAKE? Although the end-to-end (E2E) latency is reduced, readers cannot know exactly what proportion of time the distance matrix computation occupies.

4.Figure 5(a) only displays the similarity changes within extremely short steps (1540-1600 steps). This is insufficient to support the validation experiment that early FPS queries remain effective after tens of thousands of steps in ultra-long context reasoning.

5.How does this method perform on standard language modeling tasks (e.g., Needle In A Haystack (NIAH), text summarization)? There is a lack of generalization experiments.

6.Although the paper compares Max and Mean aggregation strategies in Table 2, it lacks any sensitivity analysis for the core trigger threshold $B_Q^{max}$ and downsampling target $B_Q^{min}$ in the Continual FPS algorithm. The paper only mentions in the implementation details that they are set to 32 and 16, respectively. The robustness of these two hard-coded hyperparameters across different model sizes, different context lengths, and different batch sizes remains completely unknown.

---

> ### Author Rebuttal · Authors · 2026-03-31
>
> > **W1 & Q2\.**
>
> We thank the reviewer for suggesting the "Recent + Initial" baseline. We conducted this control experiment across all models and benchmarks, and the results show that this static heuristic degrades performance on complex reasoning tasks. For example, on AIME24 with Qwen3-14B, accuracy drops from 57.92 with BeaconKV to 23.75 with Recent + Initial. This supports our key claim: reasoning models revisit multiple distinct milestones throughout long chain-of-thought, so Continual FPS is needed to preserve representative past queries beyond the prompt prefix.
>
>
> **Table R1.** Performance comparison with Recent Tokens + Initial Tokens across all models and benchmarks.
>
> | Model | Method | AIME24 | MATH-500 | GPQA-Diamond | LiveCodeBench |
> |:--|:--|--:|--:|--:|--:|
> | **R1-Distill-Qwen-7B** | BeaconKV (Ours) | 39.17 | 87.45 | 45.20 | 33.33 |
> |  | Recent + Initial | 20.42 | 70.60 | 42.80 | 28.95 |
> | **R1-Distill-Llama-8B** | BeaconKV (Ours) | 45.00 | 86.70 | 48.23 | 36.11 |
> |  | Recent + Initial | 44.58 | 81.40 | 45.07 | 33.16 |
> | **Qwen3-4B** | BeaconKV (Ours) | 33.33 | 79.75 | 50.25 | 42.21 |
> |  | Recent + Initial | 19.58 | 55.95 | 49.87 | 33.93 |
> | **Qwen3-14B** | BeaconKV (Ours) | 57.92 | 84.20 | 61.11 | 49.46 |
> |  | Recent + Initial | 23.75 | 67.30 | 60.23 | 40.23 |
>
>
> > **W2 & W3 & Q1 & Q3\.**
>
> We thank the reviewer for the opportunity to clarify efficiency. All end-to-end metrics in the paper already include the overhead of Continual FPS, including distance-matrix computation. We further compared against SnapKV and RPC in Table R2. BeaconKV increases decode latency by only about 2.5% relative to SnapKV, while improving LiveCodeBench accuracy by 11.3 points (42.2% vs. 30.9%).
>
> **Table R2. Efficiency evaluation on Qwen3-4B.**
>
> | Method   | Max KV Budget | Batch Size | Throughput (tokens/s) | Decode Latency | Peak Mem. | LiveCodeBench Acc |
> |:---------|:--------------|-----------:|----------------------:|---------------:|----------:|------------------:|
> | SnapKV   | 1K            |        320 |                1380.7 |         7594.5 |      72.4 |              30.9 |
> | RPC      | 1K            |        320 |                1380.8 |         7593.7 |      72.0 |              29.9 |
> | BeaconKV | 1K            |        320 |                1345.9 |         7790.9 |      72.5 |              42.2 |
>
> In addition, please refer to our response to Reviewer oqPg regarding the time-consumption ratio at extended sequence lengths (e.g., 32K, 64K) as requested in Q3, where we provide a detailed breakdown demonstrating that this relative overhead drops significantly, accounting for less than 3% of the total latency.
>
>
> > **W4\.**
>
> We thank the reviewer for this valuable point. While Figure 5(a) is intended to illustrate the update behavior of beacon queries rather than to serve as a long-horizon validation by itself, we expect early FPS queries to remain useful in long context reasoning because BeaconKV does not rely on a fixed set of old queries and instead updates the beacon query set online through Continual FPS. As reasoning evolves, queries that become less relevant can be replaced by more representative ones, which likely limits the impact of query aging over long decoding trajectories.
>
>
> > **W5 & Q4\.**
>
> We thank the reviewer for this suggestion. Benchmarks such as Needle-in-a-Haystack and text summarization mainly stress input-side KV during prefill, whereas our work targets long reasoning generation, where output-side KV grows during decoding and becomes the main bottleneck. More broadly, BeaconKV’s core idea may extend beyond reasoning tasks: ArkVale [1] shows that, in long-form generation, the importance of historical tokens shifts across decoding steps, so early context can become important again later. This is the kind of dynamic that BeaconKV is designed to capture, making broader long-generation settings a promising direction for future work.
>
> [1] Chen et al., ArkVale: Efficient Generative LLM Inference with Recallable Key-Value Eviction, NeurIPS 2024
>
>
> > **W6\.**
>
> We thank the reviewer for this important point. We agree that the relationship between $B_Q^{\max}$, $B_Q^{\min}$, and the recent/beacon composition should be stated more clearly. In our setting, $B_Q^{\max}=n_{\text{recent}}+n_{\text{beacon}}$ and $B_Q^{\min}=n_{\text{beacon}}$, so the current choice corresponds to $(n_{\text{recent}}, n_{\text{beacon}})=(16,16)$, i.e., $B_Q^{\max}=32$ and $B_Q^{\min}=16$. Under this parameterization, Table 1 already provides a partial sensitivity analysis, and the balanced $(16,16)$ setting gave the best overall accuracy-latency trade-off among the tested configurations. We therefore used this setting across all models and tasks, while agreeing that broader analysis across model sizes and context lengths would be valuable future work.

---

> > ### Author Rebuttal · Reviewer_JcCz · 2026-04-03
> >
> > Thanks to the authors for the responses and supplemented experiments. Regarding the response to W5, it is necessary to explicitly point out this application boundary in the "Limitations" section, which will make the paper's expression more rigorous. Regarding the response to W6, the authors did not supplement enough parameter robustness experiments. I suggest the authors integrate Table R1 and Table R2 into the main text or appendix of the final version, and according to the promise in the Rebuttal, supplement the discussion on hyperparameter sensitivity and the generalizability limitations on non-reasoning tasks.
> > I will raise my score.

---

> > > ### Author Response · Authors · 2026-04-08
> > >
> > > We sincerely thank the reviewer for the thoughtful follow-up and for taking the time to carefully reconsider our response. We are especially grateful that the reviewer found the concerns to be adequately addressed and accordingly raised the score.
> > >
> > > We also appreciate the reviewer’s recognition of the intuition behind Thought Revisiting Tokens, the strong performance of BeaconKV on heavy reasoning benchmarks under high compression, and the practical value of a training-free method without auxiliary models or fine-tuning. Our central contribution is that BeaconKV goes beyond preserving the sequence prefix: by maintaining Beacon Queries with Continual FPS, it captures dynamically revisited reasoning milestones and preserves reasoning-critical context more effectively than static heuristics such as Recent + Initial retention. In the final version, we will carefully follow the reviewer’s suggestion and explicitly state these application boundaries and limitations, as promised, to make the scope of the paper more rigorous. We will also incorporate the additional analysis from the rebuttal into the main text or appendix, and further supplement the discussion of hyperparameter sensitivity and the generalizability limitations on non-reasoning tasks. We thank the reviewer again for the constructive suggestions, which helped us improve the rigor and clarity of the paper.

---

### Official Review · Reviewer_uxDG · 2026-03-10

**Soundness:** 3
**Presentation:** 2
**Significance:** 3
**Originality:** 2
**Overall Recommendation:** 4
**Confidence:** 4

**Summary:**

BeaconKV is a training-free KV cache compression method for large reasoning models. The authors observe that certain decoding steps re-attend to distant earlier context, which clashes with recency-based eviction methods to prematurely discard important KV pairs. They find that queries associated with these revisiting steps cluster in embedding space, and propose maintaining a compact set of “beacon queries” representing those clusters to score KV entries during eviction. They implement and evaluation BeaconKV on 4 models and 4 reasoning benchmarks, showing meaningful improvements and significant memory reduction.

**Compliance With Llm Reviewing Policy:**

Affirmed.

**Key Questions For Authors:**

1. Please explain how to classify queries as global vs. local?
2. How would BeaconKV compare against simpler historical-query baselines across all benchmarks (not just AIME24 on one model)?
3. Why does Mean aggregation outperform Max at the 2048 budget in Table 2, given the claim that Max is consistently superior?
4. How sensitive is eviction quality to beacon staleness?

**Limitations:**

Yes

**Strengths And Weaknesses:**

## Strengths

S1. Well-motivated, real problem for efficient LRM deployment

S2. Compelling TRT observation about long-range attention

S3. Solid empirical evaluation

## Major weaknesses

W1. Heavy machinery for a potentially simple underlying insight

W2. Empirical validation is not fully delivered

W3. TRT definition remains somewhat vague

W4. Narrow comparison baselines

W5. Limited theoretical grounding for the clustering-based approach

W6. Paper organization blurs observation and method sections

## Detailed review

S1. Thank you for submitting this paper! It addresses a genuine and practically important problem: KV cache growth during extended chain-of-thought reasoning creates real memory bottlenecks. The problem framing is clear and well-grounded.

S2. The Thought Revisiting Tokens (TRT) observation is the strongest part of the paper. Certain decoding steps re-attend to distant earlier context, which interacts badly with recency-based KV cache eviction.

S3. The evaluation covers reasonable ground.

W1. I feel like you are introducing a lot of machinery (Continual FPS, beacon query maintenance, differential RoPE alignment, max-pooled attention scoring across GQA head groups) to express what boils down to a simple idea: score KV entries using some representative historical queries in addition to recent ones. I am not convinced all this complexity is warranted. Many of the issues identified could plausibly be addressed by improving the KV scoring function beyond pure recency, or by using sparse attention methods that already exist for long contexts. Why is this particular combination of components necessary? I am not opposed to the design, but I would like the paper to make this case clearly, which it currently fails to do. Something simpler, like incorporating attention entropy or distance-weighted importance into scoring, might get you to a similar place.

W2. The novelty hinges on three empirical claims that are not sufficiently backed up: (a) TRT is a genuinely distinct and prevalent phenomenon, not just a tail of normal attention variation; (b) query clustering is robust across layers, heads, models, and tasks; and (c) beacon selection via FPS captures future attention modes better than simpler alternatives. But the paper only shows clustering for one head of one model on one example (Figure 3). This is really thin evidence. Without broader evidence, it feels more like a well-tuned heuristic rather than a new principle.

W3. TRT is intuitively clear but operationally under-specified as you only offer a qualitative description on one example. What concrete criterion would you use to classify a query as global vs. local in practice? An attention distance threshold? An entropy cutoff? The paper does not say. I also would like to know more. For example, how frequent are TRT events across models, layers, and tasks? Are they concentrated in specific heads or layers? Currently the generality of the claims are hard to assess without broader statistics or a crisper (formal) criterion.

W4. The comparison space is too narrow. The comparison against simpler baselines exists but is insufficient and these comparisons are limited to AIME24 on one model, and the gap between FPS-selected and random beacons, while present, is only demonstrated on one benchmark and model. Several natural baselines are also missing: reservoir sampling, oldest-query retention, or clustering keys instead of queries. If the core insight is that query representativeness matters for KV scoring, I want to see these ablations across all benchmarks and models before I am convinced that Continual FPS specifically is what matters, rather than just "having some historical queries in the scoring set."

W5. The clustering argument is interesting but stays purely empirical. Why should query clusters predict future revisit patterns? When is max-pooled attention from beacon queries a reliable estimator of future token importance? What happens when future global queries fall outside the clusters observed so far? How sensitive is eviction quality to beacon staleness as reasoning progresses? The paper does not address any of these. Even an intuitive argument about when the approach is expected to work better would go a long way.

W6. The boundary between Section 3 (Observation) and Section 4 (Method) is blurry. Section 3 already introduces the idea of beacon queries as compact representatives before the method section formally proposes them. This makes it hard to tell what is an empirical finding versus a design choice. I would suggest keeping Section 3 strictly about findings (TRT exists, global queries cluster) and leaving the design proposal (FPS-selected beacons for scoring) entirely to Section 4.

(Minor) Looking at Figure 7, BeaconKV's advantage is clearest in lower budgets (256-1024 tokens). At higher budgets (2048-4096) the methods converge on some benchmarks (e.g., MATH-500), though gaps remain on harder tasks like AIME24 and LiveCodeBench. The paper should discuss this more explicitly: if the gains are primarily visible under aggressive compression, the practical value depends heavily on the target deployment scenario.

(Minor) Table 2 shows an unexpected result: at 2048 budget, Mean aggregation (50.8) actually outperforms Max aggregation (46.7). This contradicts the narrative that max-pooling is consistently superior and deserves explanation.

(Minor) The paper would benefit from stating key quantitative results in the introduction or abstract rather than relying on vague qualitative claims like "consistently outperforms."

---

> ### Author Rebuttal · Authors · 2026-03-31
>
> > **W1 & W4 & Q2\.**
>
> We thank the reviewer for asking us to more rigorously justify our design choices. Our central claim is that simply retaining a few historical queries is insufficient; what drives the performance gap in long-horizon reasoning is covering diverse, geometrically distinct query clusters. To directly test this, we compared BeaconKV against simpler baselines, including Reservoir Sampling and Oldest-Query Retention. The results in Table R1 clearly support our claim: on AIME24 with Qwen3-14B, accuracy drops from 57.92 with BeaconKV to 23.75 with Oldest-Query Retention, showing that the model revisits multiple distinct milestones rather than only the beginning of the trace. Reservoir Sampling also falls to 38.75, since random selection cannot reliably cover sparse query clusters.
>
> These results indicate that our components are not arbitrary complexity, but necessary mechanisms for capturing distinct reasoning milestones. Continual FPS improves coverage of diverse clusters under a bounded memory budget, Differential RoPE aligns historical beacons to the current decoding step, and Max-pooling preserves sparse high-magnitude signals that Mean-pooling would dilute. We will clarify the role of each component in the revised manuscript.
>
>
> **Table R1.** Performance comparison of query retention strategies across all models and benchmarks.
>
> | Model | Method | AIME24 | MATH-500 | GPQA-Diamond | LiveCodeBench |
> |:--|:--|--:|--:|--:|--:|
> | **R1-Distill-Qwen-7B** | BeaconKV (Ours) | 39.17 | 87.45 | 45.20 | 33.33 |
> |  | Reservoir Sampling | 38.33 | 84.00 | 44.57 | 33.24 |
> |  | Oldest-Query Retention | 20.42 | 70.60 | 42.80 | 28.95 |
> | **R1-Distill-Llama-8B** | BeaconKV (Ours) | 45.00 | 86.70 | 48.23 | 36.11 |
> |  | Reservoir Sampling | 41.25 | 86.15 | 45.33 | 36.47 |
> |  | Oldest-Query Retention | 44.58 | 81.40 | 45.07 | 33.16 |
> | **Qwen3-4B** | BeaconKV (Ours) | 33.33 | 79.75 | 50.25 | 42.21 |
> |  | Reservoir Sampling | 29.17 | 74.20 | 48.86 | 41.70 |
> |  | Oldest-Query Retention | 19.58 | 55.95 | 49.87 | 33.93 |
> | **Qwen3-14B** | BeaconKV (Ours) | 57.92 | 84.20 | 61.11 | 49.46 |
> |  | Reservoir Sampling | 38.75 | 81.45 | 61.75 | 47.40 |
> |  | Oldest-Query Retention | 23.75 | 67.30 | 60.23 | 40.23 |
>
>
> > **W2 & W3 & Q1\.**
>
> We thank the reviewer for requesting a more rigorous quantitative foundation for our empirical claims. To classify queries, we use an attention distance threshold. A Local Query concentrates most of its attention mass within a predefined recent window, whereas a Global Query allocates a substantial portion beyond that threshold to distant tokens. The evidence is not limited to Figure 3: Figure 1(b) shows distinct attention-distance distributions, and Figure 2 illustrates contextual revisiting. Still, we agree that broader statistical validation is needed to establish this as a stronger structural claim. In the revised manuscript, we will expand the appendix with statistical analyses of TRT occurrence frequencies.
>
> > **W5 & Q4\.**
>
> We thank the reviewer for asking for deeper intuitive grounding. Unlike ordinary conversations, extended reasoning relies on immutable early tokens such as problem constraints or initial plans. To maintain consistency, the model repeatedly returns to these premises. As a result, the global queries that induce Thought Revisiting Tokens (TRTs) do not simply become obsolete; they continue to target stable reasoning milestones, so the geometric clusters they form remain relevant throughout the trace. If reasoning enters a new phase, Continual FPS can update the beacon set online to capture new query clusters while retaining distinct older ones. In this setting, Max-pooled attention is important because TRT signals are sparse and event-driven: when a single beacon strongly attends to a critical historical token, that signal should be preserved rather than diluted by Mean-pooling.
>
>
> > **W6 & W-Minor & Q3\.**
>
> We appreciate the reviewer’s careful attention to detail. We will clarify that Max-pooling is not universally superior, but a practical choice under severe compression. When the KV budget is highly constrained (e.g., 256–512 tokens), Max-pooling helps preserve sparse critical TRT signals that Mean-pooling may dilute. This is also the regime where BeaconKV shows its clearest gains. We also thank the reviewer for the constructive suggestions on presentation. In the final version, we will restructure Section 3 to focus on empirical TRT findings and move design proposals to Section 4. We will also revise the abstract and introduction to foreground concrete quantitative results earlier in the paper.

---

> > ### Author Rebuttal · Reviewer_uxDG · 2026-04-03
> >
> > R1. The new baselines help but actually undercut the authors' narrative as Reservoir Sampling is actually competitive on several model/benchmark pairs. The strongest empirical result in the rebuttal is cherry-picked from a table that tells a much more mixed story.
> >
> > R2. The proposed attention distance threshold makes sense as possible criterion, but the authors still do not actually state it. What threshold? What percentage of attention mass? This is exactly the kind of operational specificity I asked for, but the authors punted again.
> >
> > R3. The intuitive argument about immutable early tokens is sensible and I appreciate that the authors engaged with the question.

---

> > > ### Author Response · Authors · 2026-04-08
> > >
> > > We sincerely thank the reviewer for carefully revisiting our rebuttal and for engaging further in this discussion.
> > >
> > > > **R1\.**
> > >
> > > We agree that the additional baselines present a more mixed picture than a single highlighted result might suggest. Accordingly, we believe Table R1 should be interpreted in terms of its overall pattern rather than any single model/benchmark pair such as Qwen3-14B on AIME24. The key takeaway is not that BeaconKV uniformly outperforms simpler alternatives in every case, but that it delivers a better average tradeoff across the evaluated settings, with task-dependent gains. Averaged over the four models, BeaconKV improves over Reservoir Sampling by +6.98 on AIME24, +3.07 on MATH-500, +1.07 on GPQA-Diamond, and +0.58 on LiveCodeBench. In this sense, Table R1 supports the conclusion that although simpler strategies can be competitive on some tasks, representative historical-query selection remains the stronger overall choice.
> > >
> > > > **R2\.**
> > >
> > > We agree that the operational criterion should be stated explicitly. Our threshold is a mean attention distance cutoff of 200. For a target query to be classified, we compute its attention weights over the keys available up to that step, identify the top-k most attended positions among output tokens (k = 150), measure their token distances from the query position, and take the mean of those distances. We classify the query as local if this mean distance is at most 200; otherwise, we classify it as global. This query classification, together with the geometric properties of queries observed in our analysis, provides empirical support for the design of BeaconKV. Specifically, it suggests that queries can be meaningfully grouped, which in turn motivates the clustering-based methodology underlying BeaconKV.
> > >
> > >
> > > > **R3\.**
> > >
> > > We appreciate the reviewer’s positive assessment of the intuitive grounding in our explanation. Our point is that, in extended reasoning, later steps often return to a subset of semantically stable early context, such as initial plans or key intermediate milestones, in order to maintain consistency. For this reason, the global queries associated with TRTs may remain useful over long horizons rather than becoming irrelevant after a short period. Moreover, when the reasoning trajectory enters a new phase, Continual FPS can update the beacon set online to reflect newly emerging query patterns while preserving older patterns that remain relevant.

---

### Official Review · Reviewer_oqPg · 2026-03-11

**Soundness:** 3
**Presentation:** 4
**Significance:** 4
**Originality:** 3
**Overall Recommendation:** 5
**Confidence:** 3

**Summary:**

This paper studies KV-cache compression for large reasoning models, arguing that existing eviction methods rely too heavily on recent queries as a proxy for future importance and therefore fail in long-horizon reasoning. The core observation is that some decoding steps produce Thought Revisiting Tokens (TRT) that re-attend to distant earlier context, and that the corresponding query vectors at these steps cluster into a small number of groups in query-embedding space. Based on this, the authors propose BeaconKV, a training-free compression method that maintains a compact set of representative “beacon queries” together with recent queries, so that the system can better predict which past KV pairs will matter later. They also introduce Continual Farthest Point Sampling to identify these beacon queries online with bounded memory. Empirically, across four open-source reasoning models and benchmarks spanning math, coding, and science, BeaconKV outperforms prior KV-compression baselines such as SnapKV, RPC, and R-KV, reporting up to 5.8× peak-memory reduction while nearly preserving full-cache accuracy and improving throughput by over 4.3×.

**Compliance With Llm Reviewing Policy:**

Affirmed.

**Key Questions For Authors:**

1. The method is motivated by the observation that TRT-associated global queries cluster in embedding space. How stable is this phenomenon across layers, tasks, and model scales, and are there settings where the clustering becomes too weak for beacon queries to remain effective?

2. BeaconKV appears especially well matched to long-horizon reasoning traces. How much of the gain do you expect to transfer to more general generation workloads where revisiting behavior may be less structured or less frequent?

3. The method maintains beacon queries online via Continual Farthest Point Sampling. At what compression ratios or sequence lengths does the maintenance overhead begin to offset the practical gains relative to simpler eviction heuristics?

4. The paper shows strong benchmark results, but can you characterize the main failure modes more directly? For example, when does BeaconKV most clearly break down compared with full-cache inference or even compared with simpler baselines?

5. A central claim is that recent queries are an insufficient proxy for future importance in reasoning models. Do you have evidence about whether this is primarily a property of reasoning-style decoding, or whether it should also be expected in non-reasoning LLMs?

**Limitations:**

No. The impact statement is largely positive and does not substantively discuss potential negative societal impacts or key limitations. It would be stronger to briefly acknowledge misuse risks from making long-horizon reasoning cheaper and more accessible, and to note that the empirical evidence is limited to open-source reasoning models and benchmark settings, so the gains may not generalize to broader generation workloads.

**Strengths And Weaknesses:**

**Strengths.** The paper addresses a practically important bottleneck for long-chain-of-thought inference and proposes a method that is both conceptually clean and operationally useful. The main idea is well motivated: rather than assuming recent attention is a sufficient proxy for future importance, the paper identifies a specific failure mode in reasoning models and builds a compression strategy around it. That gives the work a stronger conceptual basis than a purely heuristic eviction rule. The method is also attractive because it is training-free, appears lightweight enough for deployment, and is evaluated across multiple model families and reasoning domains. Empirically, the results look strong relative to competitive baselines, especially under aggressive compression budgets, and the gains are not limited to one benchmark or one model. Overall, this is a relevant systems contribution with a plausible mechanism and solid empirical upside.

**Weaknesses.** The main limitation is that the core mechanism rests on an empirical observation about query clustering and revisiting behavior, but the evidence for how universal and stable that phenomenon is across tasks, layers, and model scales appears more suggestive than definitive. Relatedly, the method seems tailored to the structure of long-horizon reasoning traces, so it is less clear how broadly it would generalize beyond the specific LRM setting studied here. The evaluation is fairly benchmark-driven and focuses on final accuracy, memory, and throughput; it would be useful to see more direct stress tests of failure cases, such as settings where revisiting patterns are weak, highly non-stationary, or poorly captured by a small number of beacon queries. More broadly, while the empirical gains are strong, the paper would be more convincing with a clearer accounting of when beacon-query maintenance itself becomes costly or less effective relative to simpler heuristics.

---

> ### Author Rebuttal · Authors · 2026-03-31
>
> > **Q1 & Q4\.**
>
> We thank the reviewer for raising these important questions. Empirically, we observed that the Thought Revisiting Tokens (TRT) phenomenon—and its associated query clustering—appears in multiple layers and across models of different scales. We attribute this to the nature of extended Chain-of-Thought generation, where models repeatedly re-attend to milestone tokens, such as initial problem constraints or task-solving plans, to maintain global coherence.
>
> However, while TRT itself appears consistent, the number of beacon queries needed to represent these clusters can vary by model and task. To directly examine this, we conducted an additional experiment. As shown in Table R1, when we fix the number of recent queries at 16 but reduce the beacon query budget from 16 to 4 and 1, the accuracy drops from 61.25% to 56.67% and 54.17%, respectively. In these constrained settings, the limited beacons fail to cover the necessary geometric clusters, causing BeaconKV to regress toward the simpler recency-based RPC baseline (51.25%). This result shows that sufficient beacon capacity is crucial for maintaining the method’s advantage.
>
> **Table R1. Failure case analysis under limited beacon queries on Qwen3-4B (Max KV Cache Budget = 2048).**
>
> | Method | (Recent, Beacon) | AIME24 Acc (%) |
> |:--|:--:|--:|
> | BeaconKV | (16, 16) | 61.25 |
> | BeaconKV | (16, 4) | 56.67 |
> | BeaconKV | (16, 1) | 54.17 |
> | RPC | -- | 51.25 |
>
>
>
> > **Q2\.**
>
> We agree that BeaconKV is uniquely optimized for long-horizon reasoning where Thought Revisiting Tokens (TRTs) are prominent. When revisiting behavior is less structured or less frequent, the relative performance gap between BeaconKV and simpler baselines naturally narrows. We can actually observe this tendency within our current evaluation. MATH-500 elicits the shortest generation lengths as shown in Appendix A, meaning long-horizon revisiting is inherently less critical. Consequently, as shown in Figure 7, the accuracy improvement margin of BeaconKV over RPC on Qwen3-14B for MATH-500 is somewhat modest compared to the gains seen on heavier reasoning tasks. However, BeaconKV does not deteriorate or perform worse than baselines in these general settings. Because our scoring mechanism still allocates a dedicated portion of the observation budget to recent queries, it seamlessly preserves locally relevant context just like standard recency-based methods, ensuring robust baseline performance even when global revisiting is sparse.
>
>
> > **Q3\.**
>
> Because our algorithm is computationally simple and applies Farthest Point Sampling to a strictly bounded buffer (compressing 32 queries down to 16) regardless of the overall sequence length, its absolute latency remains nearly constant at roughly 42–46 ms per 16 decoding steps. As shown in Table R2, while the relative overhead is slightly higher for very short sequences due to the weight of the initial prefill phase, it drops significantly as the sequence extends into the long-horizon regime where LRM memory bottlenecks actually occur. For generation lengths of 32K and beyond, this maintenance overhead accounts for less than 3% of the total latency. Therefore, the substantial peak memory reduction and overall throughput improvements enabled by BeaconKV's aggressive cache compression far outweigh this minimal, sub-3% maintenance cost.
>
>
> **Table R2. Total latency, FPS latency, and FPS latency overhead for 16 decoding steps across different sequence lengths on R1-Distill-Qwen-7B.**
>
> | Sequence Length | Total Latency (ms) | FPS Latency (ms) | FPS Latency Overhead (%) |
> |:--------------|------------------:|----------------:|----------------------:|
> | 128 | 921.13 | 46.14 | 5.01 |
> | 2K | 1018.13 | 42.43 | 4.17 |
> | 32K | 1478.36 | 42.59 | 2.88 |
> | 64K | 1742.98 | 42.44 | 2.44 |
>
>
>
> > **Q5\.**
>
> While our empirical focus in this work is on reasoning-style decoding, we believe the limitation of relying solely on recent queries extends to long-context generation tasks that require synthesizing or revisiting distant earlier tokens. This intuition is also supported by prior work on non-reasoning LLMs: ArkVale [1] shows that the attention score ranking of historical tokens shifts dynamically across decoding steps (Figure 4), indicating that early context can regain importance later in generation. Because BeaconKV is designed to preserve such potentially important tokens beyond strict recency, we expect it to also benefit general long-generation workloads.
>
> [1] Chen et al., ArkVale: Efficient Generative LLM Inference with Recallable Key-Value Eviction, NeurIPS 2024
>
>
> > **Limitation\.**
>
> We thank the reviewer for this valuable and constructive suggestion. In the revised manuscript, we will update our impact statement to explicitly acknowledge the potential misuse risks and properly clarify the limitations of our current empirical scope as suggested.

---

> > ### Author Rebuttal · Reviewer_oqPg · 2026-04-02
> >
> > Thank you for your response, I will maintain my positive score.

---

> > > ### Author Response · Authors · 2026-04-08
> > >
> > > We thank the reviewer for the positive overall assessment of BeaconKV, as well as for the thoughtful follow-up and encouraging feedback.
> > >
> > > We especially appreciate the reviewer’s recognition that the paper addresses a practically important bottleneck in long-chain-of-thought inference with a method that is both conceptually clean and operationally useful. Our central contribution is that BeaconKV maintains a compact set of representative beacon queries online via Continual Farthest Point Sampling under bounded memory, enabling more effective prediction of future importance than recency-based eviction alone. As clarified in the rebuttal, BeaconKV’s main failure mode arises when beacon capacity is too small to cover diverse global query patterns; we will reflect this more explicitly in the final version. In line with the reviewer’s comments on limitations, we will revise the final version to more clearly state the empirical scope of the method, explicitly acknowledge potential misuse risks, and note that the current evidence is centered on open-source reasoning models and benchmark settings. We also plan to release the BeaconKV framework to support reproducibility and facilitate further progress in this area. We thank the reviewer again for the constructive feedback, which has helped us improve the clarity and completeness of the paper.

---

### Official Review · Reviewer_Xnhw · 2026-03-12

**Soundness:** 3
**Presentation:** 3
**Significance:** 3
**Originality:** 2
**Overall Recommendation:** 5
**Confidence:** 4

**Summary:**

The paper introduces BeaconKV, a training-free Key-Value (KV) cache compression method designed specifically for Large Reasoning Models (LRMs) that generate long Chain-of-Thought (CoT) traces. The authors identify a phenomenon called Thought Revisiting Tokens (TRT): decoding steps where the model re-attends to distant, early context (like initial task plans) to maintain coherence. They observe that existing compression methods, which rely on recent queries to estimate token importance, prematurely evict these essential distant KV pairs.

Through geometric analysis, the authors find that queries inducing TRTs (global queries) cluster into small similarity groups in the embedding space. BeaconKV leverages this by maintaining beacon queries—compact representatives of these clusters—alongside recent queries to better predict future attention needs. To keep memory usage bounded, they introduce Continual Farthest Point Sampling (FPS) to efficiently update these beacons during inference. Evaluated across four LRMs (including DeepSeek-R1 distillations and Qwen3), BeaconKV achieves up to 5.8x memory reduction and 4.3x throughput improvement while maintaining near-full cache accuracy.

**Compliance With Llm Reviewing Policy:**

Affirmed.

**Final Justification:**

The rebuttal addressed my concerns and I would keep my positive score.

**Key Questions For Authors:**

Scaling Laws: How does the number of required beacon queries scale with the length of the reasoning trace? Does a 128k token trace require significantly more beacons than a 32k trace to maintain accuracy?

Model Architecture: Your results are consistent across Qwen and Llama architectures. Have you tested this on models with different attention variants, such as Sliding Window Attention (SWA), which might already "localize" attention in a way that conflicts with TRTs?

FPS Latency: While Continual FPS is memory-efficient, does the overhead of performing these geometric calculations at every eviction step impact the time-to-first-token or overall generation latency compared to simpler methods like Random or Heavy-Hitter eviction?

**Limitations:**

yes

**Strengths And Weaknesses:**

- Strengths

Soundness: The core observation of "Thought Revisiting Tokens" is well-motivated by the unique nature of LRM inference (extended CoT). The use of PCA and cosine similarity analysis to demonstrate query clustering provides a solid empirical foundation for the beacon query approach .

Significance: As LRMs move toward 32k+ token generation, the linear growth of the KV cache is a critical bottleneck. A method that reduces memory by 5.8x while preserving reasoning accuracy is highly impactful for practical deployment.

Originality: While KV eviction is a known area, the specific focus on the "global" vs "local" query distinction in reasoning traces and the proposal of "beacon queries" to represent future attention clusters is a novel contribution .


- Weaknesses

Originality/Related Work: The concept of "beacons" or "anchors" for long-context memory has some parallels in older memory-augmented neural network literature; a deeper discussion of how these "inference-time beacons" differ from training-time memory tokens would be beneficial.

---

> ### Author Rebuttal · Authors · 2026-03-31
>
> > **W1\.** Relation to Memory-Augmented Neural Network
>
> We thank the reviewer for raising this insightful connection. While both "training-time memory tokens" and our "inference-time beacons" share the goal of preserving critical context for long-horizon reasoning, their fundamental mechanisms differ significantly. Recent neural-memory approaches, such as Titans [1] and Tell Me What To Learn [2], rely on learned memory mechanisms that store or update long-term information through dedicated neural memory structures or memory-update rules, and therefore typically require training. In contrast, BeaconKV offers a highly efficient, training-free alternative where beacons act merely as pointers rather than storage. Our beacon queries do not require parameter updates; instead, they leverage the natural geometric clustering of the existing query space during inference to select and retain the most critical KV cache entries. This allows BeaconKV to effectively maintain reasoning-critical context without the prohibitive training costs or structural modifications required by learned memory modules. We will gladly expand on this distinction in the revised Related Work section.
>
> [1] Behrouz et al., Titans: Learning to Memorize at Test Time, Arxiv:2501.00663.
> [2] Bennett et al., Tell Me What To Learn: Generalizing Neural Memory to be Controllable in Natural Language, Arxiv:2602.23201.
>
>
> > **Q1\.** Scaling Laws
>
> While one might expect an extended reasoning trace to require a proportionally larger number of beacon queries, we empirically observe that the optimal number of beacons actually saturates. This saturation is a direct result of the Thought Revisiting Tokens (TRT) phenomenon : as the reasoning trace extends, the model repeatedly revisits a core set of milestone thoughts, such as initial task-solving plans or problem constraints. Because these global queries naturally fall into a small number of geometric clusters in the embedding space, a compact set of representative beacon queries remains highly effective at covering the critical key-values, regardless of the overall sequence length. Furthermore, under a fixed observation budget, over-allocating to beacon queries restricts the capacity for recent queries, which degrades local reasoning coherence (as shown in Table 1). Therefore, an extended reasoning trace does not require a proportional increase in beacons to maintain accuracy.
>
> > **Q2\.** Model Architecture
>
> We thank the reviewer for highlighting the interesting intersection between BeaconKV and attention variants like Sliding Window Attention (SWA). As the reviewer points out, SWA strictly enforces locality by retaining only recent KV pairs, which inherently prevents the emergence of Thought Revisiting Tokens (TRTs); recent studies such as SWAA [3] and SWAT [4] likewise suggest that relying solely on SWA is disadvantageous for long-horizon reasoning because it blocks re-attention to distant critical context. To address this limitation, recent state-of-the-art models (e.g., GPT-OSS, Gemma 3) increasingly adopt hybrid architectures that interleave SWA layers with periodic Full-KV attention layers. However, the memory overhead of maintaining these Full-KV layers during extended generation remains substantial. We therefore view BeaconKV as a promising complementary direction: rather than conflicting with SWA, it can be selectively applied only to the Full-KV attention layers in such hybrid models, preserving the global TRT pathways needed for complex reasoning while further improving memory efficiency.
>
> [3] Yu et al., SWAA: Sliding Window Attention Adaptation for Efficient Long-Context LLMs Without Pretraining. Arxiv:2512.10411.
> [4] Fu et al., Sliding Window Attention Training for Efficient Large Language Models. Arxiv:2502.18845.
>
>
> > **Q3\.** FPS Latency
>
> We appreciate the reviewer's careful consideration of computational overhead. Although calculating the initial geometric clusters for Continual FPS slightly increases the Time-to-First-Token (TTFT) during prefill, overall inference time for Large Reasoning Models is dominated by the extended decoding phase. During generation, Continual FPS enables more aggressive KV cache compression without losing critical context, and its geometric calculations are performed only once every 16 steps, when the query buffer reaches 32 and is downsampled to 16. The Decode Latency metrics reported in Table 3 already include all overhead associated with maintaining and updating these beacons. Compared directly with RPC—a simpler recency-based eviction baseline that requires no clustering calculations—BeaconKV increases decoding latency by less than 3% (e.g., 8926.9s vs. 8672.5s at a 2K budget). Given the substantial reasoning accuracy gains over RPC across multiple benchmarks (Figure 7), we believe this sub-3% overhead is a favorable practical trade-off. Please see our response to Reviewer oqPg for FPS time consumption at various sequence lengths.

---

> > ### Author Rebuttal · Reviewer_Xnhw · 2026-04-01
> >
> > My concerns have been addressed

---

> > > ### Author Response · Authors · 2026-04-08
> > >
> > > We thank the reviewer for the positive assessment of our work and for recognizing the soundness, significance, and originality of BeaconKV.
> > >
> > > In particular, we appreciate the reviewer’s recognition of the importance of Thought Revisiting Tokens (TRT), the practical value of training-free KV cache compression for Large Reasoning Models with long Chain-of-Thought traces, and the novelty of beacon queries as compact representatives of future attention clusters. Our core contribution is to show that inference-time beacons, maintained via Continual FPS, can effectively preserve reasoning-critical distant context under bounded memory, providing a practical alternative to recency-based eviction for long-horizon reasoning. As we clarified in the rebuttal, this mechanism is fundamentally different from the training-time memory tokens used in memory-augmented neural networks, since beacon queries act as pointers rather than storage and require neither additional training nor architectural modification. In the final version, we will make this distinction clearer in the Related Work section. We thank the reviewer again for the thoughtful feedback and for engaging constructively with our rebuttal.

---

### Decision · Program_Chairs · 2026-04-30

**Decision:**

Accept (regular)

**Comment:**

This paper introduces BeaconKV, a training-free KV cache compression method tailored for Large Reasoning Models (LRMs) that generate extended Chain-of-Thought traces. The authors identify that certain decoding steps, termed Thought Revisiting Tokens (TRT), re-attend to distant historical contexts rather than recent ones, causing traditional recency-based eviction methods to fail. To address this, the proposed method utilizes Continual Farthest Point Sampling to maintain a compact set of representative "beacon queries" from historical clusters, allowing the model to better predict and preserve crucial KV pairs for future attention, thereby significantly reducing memory overhead and improving throughput.

The reviewers unanimously praised the paper's strong motivation and the practical significance of the addressed problem, particularly as extended reasoning traces increasingly bottleneck LRM deployment [Reviewer Xnhw, Reviewer oqPg, Reviewer uxDG, Reviewer JcCz]. The core observation regarding TRTs provides a compelling and conceptually clean foundation that aligns well with current intuitions about reasoning mechanisms [Reviewer oqPg, Reviewer uxDG, Reviewer JcCz]. Furthermore, the proposed training-free approach demonstrates impressive empirical performance across multiple benchmarks and model architectures, achieving substantial memory reductions while preserving high accuracy compared to existing baselines [Reviewer Xnhw, Reviewer oqPg, Reviewer JcCz].

Despite its strengths, reviewers initially raised several concerns regarding the method's complexity, the generalizability of the TRT clustering phenomenon, and potential computational overhead [Reviewer oqPg, Reviewer uxDG, Reviewer JcCz]. Specifically, reviewers questioned whether simpler heuristics could achieve similar results [Reviewer uxDG, Reviewer JcCz] and noted a lack of diverse baselines and generalization tests on standard NLP tasks [Reviewer uxDG, Reviewer JcCz]. During the rebuttal, the authors successfully addressed concerns by providing additional experimental data that justified the necessity of the Continual FPS mechanism over static baselines and demonstrated a favorable accuracy-latency trade-off [Reviewer JcCz]. The authors also committed to explicitly detailing the application boundaries and parameter robustness in the final manuscript [Reviewer JcCz].

Given that the authors adequately resolved the most critical concerns regarding overhead and baseline comparisons during the rebuttal phase, all reviewers agrees that the contributions are significant and timely. Therefore, I recommend that this paper be accepted.